# RA-CoA: Training-free Fashion Image Captioning via Retrieval-Augmented Chain-of-Attributes

**Abhirama Subramanyam Penamakuri**[*]                        *venkata.penamakuri@mbzuai.ac.ae*
*Department of Natural Language Processing*
*Mohamed Bin Zayed University of Artificial Intelligence*

**Shreya Shukla**[*]                                         *shreyashukla628@gmail.com*

**Anand Mishra**                                            *mishra@iitj.ac.in*
*Department of Computer Science and Engineering*
*Indian Institute of Technology Jodhpur*

**Reviewed on OpenReview:** `https://openreview.net/forum?id=PpkOrVUpJ6`

## Abstract

Fashion Image Captioning (FIC) plays a vital role in enhancing user experience and product search in e-commerce platforms. Unlike natural scene image captioning, FIC requires fine-grained visual reasoning and knowledge of domain-specific terminology to capture subtle attributes such as neckline and closure types, graphic patterns, and dress silhouettes. Moreover, as fashion inventories evolve rapidly with new trends, styles, and frequently emerging vocabulary, developing training-free captioning solution becomes essential for scalability and real-world adaptability. Instruction-tuned vision-language models (VLMs) offer a promising solution to fashion image captioning due to their strong zero-shot capabilities and natural language fluency. However, these general-purpose models often lack attribute-level coverage and precision, and tend to hallucinate or misidentify fine-grained fashion details, making them less suitable for high-fidelity applications like product cataloging or personalized recommendations. To address this, we propose RA-CoA (**R**etrieval-**A**ugmented **C**hain-of-**A**ttributes), a novel, training-free framework that disentangles fashion image captioning into two interpretable stages: (i) retrieval of relevant attribute sets from a product knowledge base, and (ii) attribute-level reasoning to generate the final caption. RA-CoA is a model-agnostic approach that works with frozen VLMs to improve fine-grained attribute precision in product captions without the need for fine-tuning. Extensive evaluations across diverse VLM model families under different prompting paradigms demonstrate that RA-CoA significantly improves caption quality, achieving an average gain of 26.3% METEOR score over zero-shot captioning. We make our code publicly available[1].

## 1 Introduction

With the rise of AI-driven personalization in e-commerce, Fashion Image Captioning (FIC) has emerged as a crucial task for enhancing user experience and improving product searchability[2]. Unlike the well-established task of natural scene image captioning Hodosh et al. (2013); Lin et al. (2014); Krishna et al. (2017), fashion image captioning (FIC) Yang et al. (2020); Rostamzadeh et al. (2018) necessitates *fine-grained visual reasoning*

---

[*]Equal contribution and work done while at Indian Institute of Technology Jodhpur.

[1]`https://github.com/vl2g/RACoA`

[2]According to Google consumer research, approximately 85% of shoppers report that accurate product information play an important role in deciding which brand or retailer to buy from, highlighting the need for accurate attribute-level product descriptions in e-commerce Google (2019).

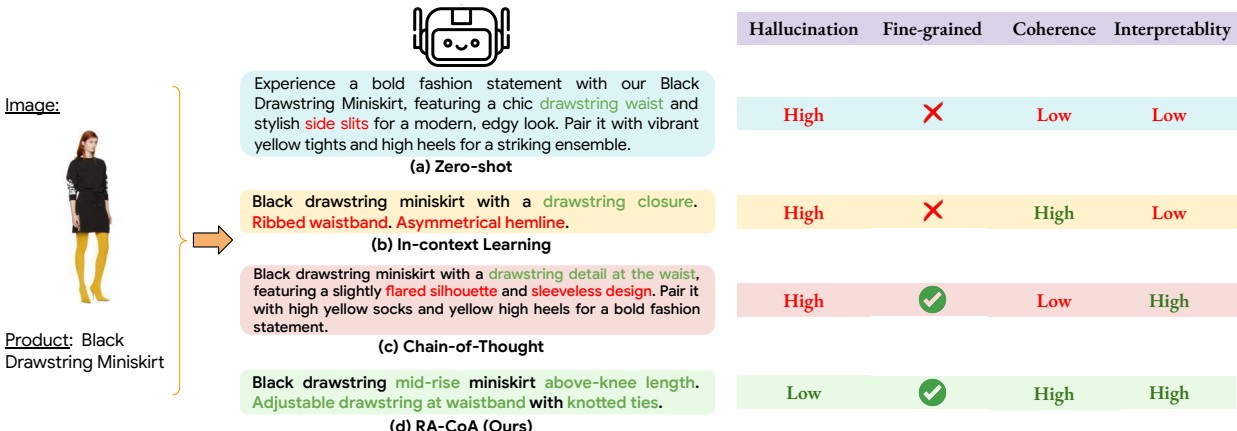

Figure 1: **Task Overview:** Given a fashion image and the product name, existing training-free prompting paradigms may fail to provide attribute-rich, coherent, hallucination-free captions. Captioning with (a) Zero-shot: Produces only a generic description, missing key fine-grained attributes. (b) In-context learning Dong et al. (2022): Improves coherence via exemplars but tends to copy attributes. (c) Chain-of-Thought Wu et al. (2025): Performs stepwise reasoning yet fails in capturing all fine-grained attributes correctly. (d) RA-CoA (Ours): A retrieval-augmented chain-of-attributes framework that aggregates key attributes from similar catalog items and sequentially predicts their values. The predicted attributes, combined with ICL exemplars, guide caption generation toward coherent and attribute-faithful captions.

to accurately capture nuanced product attributes such as stylistic patterns, closure mechanisms, collar and sleeve silhouettes, etc. Furthermore, the rapidly evolving nature of fashion inventories demands *comprehensive knowledge of domain-specific terminology*, including emerging trends, seasonal styles, and an ever-expanding attribute vocabulary. In this dynamic landscape, curating annotated datasets and retraining domain-specific supervised captioning models from time-to-time becomes prohibitively expensive and operationally impractical. This challenge highlights the critical need for training-free frameworks that deliver adaptability, scalability, and efficiency while maintaining high-quality product descriptions.

Instruction-tuned vision-language models (VLMs) Liu et al. (2024a); Zhu et al. (2023); Ye et al. (2023); Liu et al. (2024b); Bai et al. (2025); Chen et al. (2024); OpenAI (2024) present a compelling solution. Their capacity to generate fluent, naturalistic descriptions without task-specific training makes them attractive candidates for fashion image captioning. However, these general-purpose models fundamentally lack the attribute-level precision required for fashion domains, frequently hallucinating or misidentifying fine-grained attributes critical to product identity. Consequently, their zero-shot outputs remain inadequate for e-commerce applications that demand accurate and faithful descriptions for automated product cataloging, fine-grained fashion retrieval, and personalized recommendation systems.

Recent advances demonstrate that vision-language models can benefit substantially from structured prompting strategies such as in-context learning Dong et al. (2022); He et al. (2025) and chain-of-thought reasoning Wu et al. (2025). These techniques have proven effective in eliciting more accurate and coherent responses from VLMs across various tasks. However, in the fashion domain, even these advanced prompting strategies struggle to ground fine-grained attributes reliably, as illustrated in Figure 1. *In-context learning*, while effective at improving stylistic coherence through exemplar-based prompting, suffers from attribute hallucination. Models tend to erroneously copy or interpolate attributes from provided examples rather than faithfully grounding them in the target image. *Chain-of-thought prompting*, despite enabling stepwise reasoning, still struggles with comprehensive fine-grained attribute identification, as VLMs lack the visual acuity to reliably distinguish subtle fashion-specific details (e.g., differentiating between a mandarin collar and a band collar) when reasoning from scratch. The core challenge lies in the fact that directly querying VLMs for fine-grained attributes, often results in hallucinated or imprecise responses due to insufficient visual grounding and the absence of domain-specific knowledge.

To address these limitations, we propose RA-CoA (**R**etrieval **A**ugmented **C**hain **o**f **A**ttributes), a training-free framework that synergistically combines the zero-shot reasoning capabilities of VLMs with explicit retrieval-based grounding and structured attribute reasoning. Unlike direct caption generation or conventional prompting approaches, RA-CoA decomposes the captioning task into two interpretable and complementary stages: (i) *Attribute Set Retrieval*, where we leverage a product knowledge base (ProductKB) to retrieve product-aware visually similar examples and extract candidate attributes; and (ii) *Chain-of-Attribute Reasoning*, where the model performs attribute-by-attribute visual grounding, inferring the value of each candidate attribute from the image, before synthesizing these verified attributes into a coherent, factually grounded caption. By introducing retrieval-augmented attributes prior to reasoning, RA-CoA mitigates the hallucination problems inherent in in-context learning (by providing explicit attribute candidates rather than exemplar captions to copy from) and chain-of-thought (by constraining the reasoning space to visually plausible attributes). This explicit decomposition makes fine-grained visual reasoning interpretable and verifiable, enabling the model to focus its attention on relevant visual regions for each attribute. Importantly, RA-CoA is modular, model-agnostic, and operates with frozen VLMs, requiring no fine-tuning which enables seamless adaptation to evolving fashion inventories and emerging attribute taxonomies. Through this structured retrieval-augmented approach, RA-CoA achieves a balance between zero-shot flexibility and fine-grained precision, producing accurate and interpretable captions with minimal hallucination.

The key contributions of our work are threefold:

1. We propose RA-CoA, a novel *training-free* and *model-agnostic* framework for fashion image captioning that enhances fine-grained visual grounding by decomposing caption generation into retrieval-augmented attribute identification followed by explicit chain-of-attribute reasoning.

2. We compare RA-CoA against a range of prompting paradigms including vanilla zero-shot, in-context learning, implicit and explicit chain-of-thought prompting, across both open and closed-source VLMs. Through automatic metrics and user studies, we show that RA-CoA consistently improves attribute precision and semantic coherence in generated captions compared to baselines.

3. We demonstrate the robustness and real-world applicability of RA-CoA by benchmarking it against a state-of-the-art supervised fashion captioning model on a held-out, web-curated data. Despite operating without any task-specific training or fine-tuning, RA-CoA achieves superior performance over the state-of-the-art method, highlighting its ability to generalize under distribution shift and its suitability for scalable deployment in dynamic e-commerce settings.

## 2 Related Work

### 2.1 Fashion image captioning

Fashion image captioning has emerged as a specialized task within the broader field of vision-and-language understanding, and presents unique challenges requiring models to generate accurate, attribute-rich captions that capture both visual appearance and semantic fashion concepts. Early work in this domain focused on fashion retrieval and attribute prediction with datasets such as DeepFashion Liu et al. (2016), Fashion200K Han et al. (2017) and FasionIQ Wu et al. (2021). These datasets provide rich annotations for fashion-specific attributes including garment categories, colors, patterns, and styles. However, these datasets are designed for retrieval-centric tasks with relative captions, making them less suited for evaluating free-form captioning. More recent approaches have explored fashion image captioning with FashionGen Rostamzadeh et al. (2018) dataset, which offers comprehensive, attribute-rich product descriptions specifically designed for caption generation. Several models have been developed for this task. Li et al. (2021) pioneered the integration of explicit attribute detection with visual attention mechanisms. The recent rise of large-scale vision-language pre-training has significantly advanced the field with specialized fashion models. FashionBERT Gao et al. (2020) adapts masked language modeling for fashion attribute prediction. KaleidoBERT Zhuge et al. (2021) introduces adaptive fashion-text pre-training with dynamic masking strategies. Methods like FashionVLP Goenka et al. (2022) leverage vision-language pre-training specifically adapted for fashion domains to improve attribute-aware description generation. FashionViL Han et al. (2022) proposes contrastive learning between fashion

images and text descriptions. FAME-ViL Han et al. (2023) incorporates multi-modal entity understanding for fashion, and UniFashion Zhao et al. (2024) presents a unified framework for multiple fashion tasks including captioning and retrieval. However, most existing approaches require substantial task-specific fine-tuning on fashion datasets, creating significant computational overhead and limiting their adaptability to the evolving vocabulary and products in the catalogs. In contrast to these training-intensive paradigms, our proposed RA-CoA is a novel model-agnostic training-free approach that leverages retrieval-augmented structured reasoning to generate attribute-grounded captions in a zero-shot setting.

## 2.2 Retrieval-augmented image captioning

Retrieval-Augmented Generation (RAG) was initially proposed to ground language generation in external knowledge, reducing reliance on parametric memory and improving factuality Lewis et al. (2020). This idea has since been adopted in image captioning through retrieval-augmented *training*, where models learn to condition caption generation on retrieved samples. Early retrieval-augmented image captioning models such as EXTRA Ramos et al. (2023a) jointly encode images and retrieved captions to learn retrieval-conditioned generation. Subsequent analysis has shown that such models are sensitive to retrieval noise and the ordering of retrieved samples Li et al. (2024b). SmallCap Ramos et al. (2023b) explores a lighter alternative by prompting a largely frozen language model with retrieved captions and training only lightweight components to internalize retrieval signals. EVCap Li et al. (2024a) further extends this training-based paradigm by introducing an external visual–name memory to support open-world captioning. More recently, retrieval has also been explored for zero-shot captioning. Visual RAG Bonomo & Bianco (2025) proposes CLIP-based retrieval-augmented in-context learning for image classification, though its applicability to fine-grained, open-ended attribute-level caption generation remains limited. MeaCap Zeng et al. (2024) addresses open-domain zero-shot image captioning by retrieving textual memory and extracting salient concepts to guide generation. This formulation emphasizes concept completion and semantic plausibility in natural scene images, rather than faithful prediction of structured visual attributes, which is crucial for domains such as fashion. In contrast to these works, RA-CoA targets attribute-faithful captioning in structured domains and is *training-free* by design. It operates in a zero-shot setting using off-the-shelf vision–language models, without any fine-tuning or task-specific supervision. Rather than conditioning generation on retrieved captions or attribute values, RA-CoA retrieves only attribute keys and infers attribute values directly from the VLM's internal knowledge through structured reasoning.

## 2.3 Zero-shot prompting strategies for Vision Language Models

Large-scale vision-language models (VLMs) Li et al. (2023); Zhu et al. (2023); Ye et al. (2023); Liu et al. (2024a); Dai et al. (2024); Zhou et al. (2024); Bai et al. (2025); Chen et al. (2024); OpenAI (2024) have demonstrated strong performance transfer across diverse tasks. Zero-shot learning paradigm with VLMs leverages pre-trained knowledge without task-specific fine-tuning. Beyond basic zero-shot prompting, several advanced prompting paradigms have emerged to elicit optimal performance from VLMs. In-context learning (ICL) Dong et al. (2022); He et al. (2025) extends zero-shot capabilities by providing few-shot examples as demonstrations, enabling models to adapt to new tasks without parameter updates. Chain-of-thought (CoT) Wei et al. (2022); Wu et al. (2025) prompting decomposes complex reasoning into intermediate steps, initially proposed for language models and later adapted to vision-language tasks. However, when applied to the task of fashion image captioning, current zero-shot prompting strategies struggle with attribute-level precision, often generating generic descriptions that miss intricate design elements of the fashion product. In-context learning frequently suffers from memorization bias Golchin et al. (2024), where models reproduce content from reference samples rather than adapting to the specific visual attributes of the target image. Chain-of-thought prompting shows limited effectiveness in attribute-centric captioning scenarios where structured domain knowledge and systematic visual analysis are required. These limitations underscore the need for more sophisticated prompting paradigms that can effectively guide VLMs to generate detailed and accurate captions for fashion products. Our work addresses this gap by combining retrieval with explicit attribute reasoning to guide zero-shot captioning in VLMs.

## 3 Methodology

In this section, we present RA-CoA (**R**etrieval **A**ugmented **C**hain **o**f **A**ttributes), a novel training-free approach for fashion image captioning (FIC). RA-CoA works in a step-wise manner by first identifying attributes, then assigning values, and finally composing a coherent description. This stepwise attribute-centric reasoning enables fine-grained understanding prior to generating captions.

**Problem Statement:** We define fashion image captioning as an attribute-driven process. Given an image of a fashion product **I** and its name **N**, the goal is to generate a caption **C** using a set of attribute-value pairs $\mathcal{A} = \{(a_i, v_i)\}_{i=1}^{n}$ for the specific product, where $a_i$ denotes an attribute type, e.g., sleeve type, neckline and $v_i$ its corresponding value, e.g., half-sleeve, v-neck. RA-CoA models this in three structured stages: **(i) Identification of attributes**: Determine the relevant attributes $\{a_1, a_2, \cdots, a_n\}$ for the product image **I**, **(ii) Assignment of values**: For each attribute $a_i$, reasoning about its appropriate value $v_i$, and **(iii) Caption generation**: Synthesizing caption $C_t$ by integrating these attribute-value insights. This reasoning-based decomposition enables fine-grained visual attribute reasoning for fashion image captioning.

### 3.1 ProductKB Construction

We construct a structured Product Knowledge Base (ProductKB) of product images, their captions, and attribute-value tabular data to support retrieval-augmented generation. We leverage the training dataset of FashionGen Rostamzadeh et al. (2018), $\mathcal{D} = \{(I_j, N_j, C_j)\}_{j=1}^{m}$ of $m$ triplets for this purpose. Each triplet consists of a fashion product image ($I_j$), product name ($N_j$) and its caption ($C_j$). Next, we extract structured attribute-value pairs from the captions using the Llama-3.2-3B-Instruct model ($M$) Grattafiori et al. (2024): $M_{LLM}(C_j) \rightarrow \mathcal{A}_j = \{(a_j, v_j)\}_{i=1}^{m}$, guided by this prompt:

---

**Prompt used for ProductKB construction**

You are given features $\{C_j\}$ for an e-commerce product, identify the keys for these features and output it in json format. Keys should be very short, precise and specific, such that if given only the key and product image, its feature value can be predicted. Keys should cover all the features of the product. Break down the features into multiple key-value pairs where appropriate. Do not create sub-keys or sub-values. Both key and value should be str datatype. Strictly output only the json.
Assistant: $\{\mathcal{A}_j\}$.

---

Popular fashion datasets such as FashionGen Rostamzadeh et al. (2018) often depict models wearing multiple products within a single image, which introduces ambiguity when associating captions and attributes with a specific product. To ensure that each entry in the ProductKB corresponds precisely to the intended product, we leverage the product name $N_j$ to isolate the target product from $I_j$. Specifically, we use the Florence-2 Xiao et al. (2024) model ($f$) to detect and crop the image region corresponding to the product, resulting in a focused crop: $I_j^c = f(I_j, N_j)$. This step ensures that the stored representation in ProductKB is accurately aligned with the product described in the accompanying caption and attribute annotations.

Our final ProductKB ($\mathcal{P}$) comprises of $\{(I_j^c, N_j, C_j, \mathcal{A}_j)\}_{j=1}^{m}$, where each entry includes the cropped image focused on the target product $I_j^c$, the product name $N_j$, the expert-written caption $C_j$, and the obtained structured attribute-value pairs $\mathcal{A}_j$. This curated knowledge base supports the efficient retrieval of visually similar products with shared attribute schemas, facilitating accurate and attribute-based caption generation.

### 3.2 RA-CoA: **R**etrieval-**A**ugmented **C**hain-**of**-**A**ttributes

RA-CoA integrates a retrieval mechanism with a structured reasoning chain for fashion image captioning. The pipeline proceeds in multiple steps: retrieving visually similar products, identifying relevant attribute keys, estimating their values, and finally composing a caption grounded in these attributes. The following subsections elaborate on each stage of the pipeline.

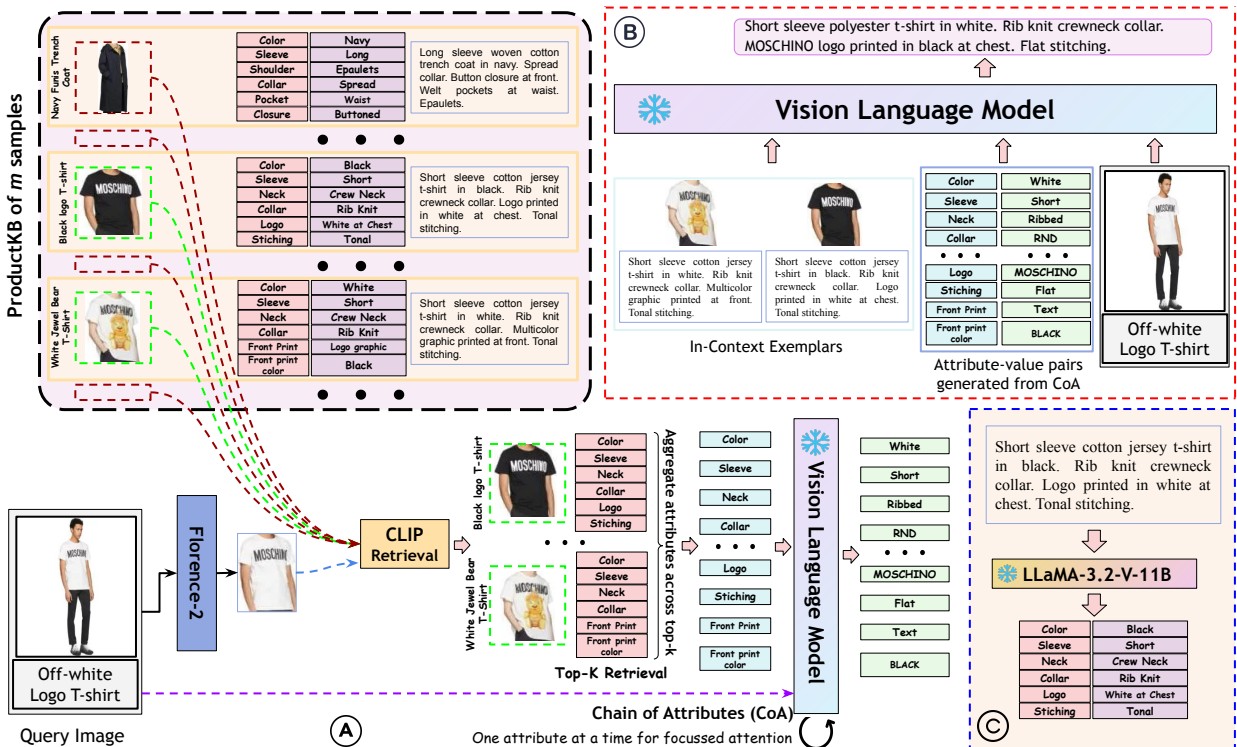

Figure 2: **Overview of RA-CoA.** Ⓐ *Retrieval-augmented Chain-of-Attributes* - For a query fashion image, we crop the target product region using Florence-2, retrieve top-K similar products from ProductKB, and aggregate unique attributes. Each attribute is paired with the image and product name and fed to VLM for value prediction. Ⓑ *Caption Generation* - The resulting attribute-value pairs and in-context exemplars (retrieved products) enable the VLM to generate fine-grained coherent captions. Ⓒ *ProductKB Construction* - We construct our ProductKB by extracting structured attribute-value pairs from FashionGen captions using LLaMA-3.2-V-11B.

### 3.2.1 Image embedding and retrieval

The first step involves retrieving relevant attribute knowledge from the ProductKB based on visual similarity. Given a query image $\mathbf{I}$ and its associated product name $\mathbf{N}$, we use the Florence-2 Xiao et al. (2024) model ($f$) to isolate the target product and obtain the cropped image $\mathbf{I}^c = f(\mathbf{I}, \mathbf{N})$. We then compute its visual embedding $e$ using a pre-trained CLIP Radford et al. (2021) vision encoder ($E_V$), i.e., $e = E_V(\mathbf{I}^c)$. In a similar manner, we encode all product images in the ProductKB using the same encoder and retrieve the top-$K$ most similar items based on cosine similarity. The resulting set is denoted as $\mathcal{P}_K = \{(I_i^c, N_i, C_i, \mathcal{A}_i)\}_{i=1}^K$. While attributes $\{\mathcal{A}_i\}_{i=1}^K$ are used to guide value prediction, $\{(I_i^c, N_i, C_i)\}_{i=1}^{K_{icl}}$ are used later in the pipeline as in-context exemplars during caption generation.

### 3.2.2 Attribute extraction

Once top-K similar products are retrieved from the ProductKB, we extract and aggregate the unique attributes (Eq. 1).

$$\mathcal{K} = \bigcup_{i=1}^{K} a | (a, v) \in \mathcal{A}_i \qquad (1)$$

We extract only attribute keys ($\mathcal{K}$), excluding their values, as similar products often share attribute types (e.g., sleeve type, neckline) but differ in specifics. This avoids propagating incorrect values while retaining guidance on which attributes to predict, ensuring value inference remains grounded in the query image.

---

**Algorithm 1** RA-CoA Pipeline

---

**Input:** Query image $\mathbf{I}$, Query product name $\mathbf{N}$, ProductKB: $\mathcal{P}$ and Vision-Language Model $M_{VLM}$.
**Output:** Generated caption $\tilde{\mathbf{C}}$

1: $\mathbf{I}^c = f(\mathbf{I}, \mathbf{N})$        ▷ Get product-of-interest crop.
2: $e \leftarrow E_V(\mathbf{I}^c)$        ▷ Generate query embedding.
3: $\mathcal{P}_K \leftarrow$ Retrieve top-$K$ similar products from $\mathcal{P}$ using $sim(e, e_j)_{j=1}^m$.
4: $\mathcal{P}_K = \{(I_j^c, N_j, C_j, \mathcal{A}_j)\}_{j=1}^K$      ▷ Each retrieved product has image, name, caption and attributes.
     $\mathcal{E}_{K_{icl}}$ used during caption generation is a subset of $\mathcal{P}_K$
5: $\mathcal{K} \leftarrow \bigcup_{(I_j^c, N_j, C_j, \mathcal{A}_j) \in \mathcal{P}_K} \{a \mid (a, v) \in \mathcal{A}_i\}$        ▷ Extract unique keys
6: $\tilde{\mathcal{A}} \leftarrow \emptyset$        ▷ Initialize attribute set for query image.
7: **for** each $a_j \in \mathcal{K}$ **do**
8:    $v_j \leftarrow M_{VLM}(p_{value}(\mathbf{I}, \mathbf{N}, a_j))$        ▷ Generate attribute value.
9:    $\tilde{\mathcal{A}} \leftarrow \tilde{\mathcal{A}} \cup \{(a_j, v_j)\}$        ▷ Add to attribute set.
10: **end for**
11: $\tilde{\mathbf{C}} \leftarrow M_{VLM}(p_{caption}(\mathbf{I}, \mathbf{N}, \tilde{\mathcal{A}}, \mathcal{E}_{K_{icl}}))$        ▷ Generate final caption.
12: **return** $\tilde{\mathbf{C}}$

---

### 3.2.3 Chain-of-Attributes (CoA)

Once the relevant attributes $\mathcal{K} = \{a_1, a_2, ..., a_n\}$ are identified through retrieval, we implement a Chain-of-Attributes reasoning process. Analogous to chain-of-thought reasoning, CoA decomposes fashion understanding into attribute-wise focus steps. For each attribute $a_i \in \mathcal{K}$, we query the VLM to determine its corresponding value:

$$v_i = M_{VLM}(p_{value}(\mathbf{I}, \mathbf{N}, a_i)) \tag{2}$$

where $p_{value}(.)$ is a prompt template for attribute value generation:

> **Prompt used for attribute value generation**
>
> $< image\ (\mathbf{I}) >$
> Given the image of a model wearing the e-commerce product - $\mathbf{N}$, how/what is the $a_i$ of the product? Strictly answer as single word or phrase.
> Assistant: $\{v_i\}$.

This process results in a structured attribute-value set $\tilde{\mathcal{A}} = \{(a_1, v_1), (a_2, v_2), ..., (a_n, v_n)\}$ where each attribute receives focused attention. By isolating individual attribute queries, CoA reduces ambiguity and cognitive load, leading to more accurate and interpretable value predictions.

### 3.2.4 Caption generation

Once we obtain the comprehensive set of attribute-value pairs $\tilde{\mathcal{A}}$, the final caption is generated by conditioning the vision-language model on the original query image $\mathbf{I}$, product name $\mathbf{N}$, and the top-$K_{icl}$ retrieved examples $\mathcal{E}_{K_{icl}} = \{(I_i^c, N_i, C_i)\}_{i=1}^{K_{icl}}$. These retrieved entries serve as in-context exemplars to guide both the structure and style of the generated description. The caption is produced as:

$$\tilde{\mathbf{C}} = M_{VLM}(p_{caption}(\mathbf{I}, \mathbf{N}, \tilde{\mathcal{A}}, \mathcal{E}_{K_{icl}})). \tag{3}$$

where $p_{caption}(.)$ is a prompt template for caption generation as shown below:

---

**Prompt used for caption generation**

$<image \ (\mathbf{I}) >$
Given the image of a model wearing the e-commerce product - $\mathbf{N}$, and the attribute-value pairs of the product - $\tilde{\mathcal{A}}$, write a concise caption for the specific product, incorporating its fine-grained attributes. Take reference from the provided examples:
Example1: $<image \ (I_1^c)>$ Product name: $N_1$, Caption: $C_1$
Example2: $<image \ (I_2^c)>$Product name: $N_2$, Caption: $C_2$
Output only the final caption.
Assistant: $\tilde{\mathbf{C}}$

---

This final synthesis step integrates the individually analyzed attributes into a holistic description of the fashion item. The complete RA-CoA pipeline is summarized in Algorithm 1.

## 4 Experiments and Results

### 4.1 Dataset

We conduct our experiments on the FashionGen Rostamzadeh et al. (2018) dataset, which contains roughly 300K images of e-commerce products in the fashion domain, each accompanied by rich metadata and expert-written descriptions that capture fine-grained attributes of the product. In our work, we leverage only three fields: product images, names/titles, and ground-truth captions. FashionGen images often feature a model wearing multiple fashion product - clothing accessorized with bags, shoes, or jewelry. Hence, the target product may not be immediately obvious. Thus, we utilize product names as a guiding cue to focus the VLM's attention towards the intended item. The expert-written descriptions then serve as ground-truth captions for evaluation. From the FashionGen training split, we sample 60K products to build our retrieval knowledge base (KB), which the RA-CoA module uses exclusively for product name-aware image retrieval; these KB samples are never used to train or validate the VLMs. We then assessed the captioning performance on the 7K test samples, comparing the generated captions directly against the FashionGen captions.

### 4.2 VLMs used

We evaluate a diverse set of state-of-the-art vision–language models to benchmark the effectiveness of our approach across varying model scales and capabilities. To ensure comprehensive coverage, we include (a) *Small VLMs (models with parameters <4B)*: (i) TinyLLaVA (3B) Zhou et al. (2024), and (ii) Qwen-2.5VL (3B) Bai et al. (2025) and (b) *Large VLMs (models with parameters >4B)*: (i) Qwen-2.5VL (7B) and (ii) InternVL2 (8B) Chen et al. (2024), and (c) *Closed-source model*: GPT-4o OpenAI (2024).

#### 4.2.1 VLM Paradigms

We compare our proposed framework RA-CoA against standard VLM prompting techniques such as **(i) Direct prompting** (Zero-shot): where VLM is prompted with an instruction to generate the description directly given an image. **(ii) Few-shot prompting** (In-context Learning - ICL): Here, we follow the standard few-shot in-context prompting technique, where we retrieve $K_{icl}$ random product images along with their tabular information, and add them to the prompt as in-context exemplars. **(iii) Implicit Chain-of-thought** (CoT-i) prompting: We prepend a generic "let's think step by step" cue to the zero-shot prompt, encouraging the VLM to internally structure its reasoning to identify fine-grained attribute-value pairs, without exposing intermediate steps in the output., **(iv) Explicit Chain-of-Thought** (CoT-e): Here, we explicitly ask the model to enumerate each reasoning step, itemizing attributes one by one before producing the final caption, thereby surfacing its internal chain of thought., **(v) Explicit Chain-of-Thought with few-shot prompting** (ICL+CoT-e): This constitutes a stronger baseline by combining advantages of (ii) and (iv). The detailed prompts used for each of these settings are provided below.

### 4.3 Prompts used for different paradigms

In this section, we enlist the prompts used for various VLMs under different training-free paradigms used as baselines. These prompts remain consistent across all VLMs including the closed-source GPT-4o.

---

**Prompt used under zero-shot paradigm**

<image($\mathbf{I}$)>
Given the image of a model wearing the e-commerce product - $\mathbf{N}$, write a concise caption for the specific product, incorporating its fine-grained attributes.
Assistant: $\tilde{\mathbf{C}}$

---

**Prompt used under implicit CoT paradigm**

<image($\mathbf{I}$)>
Given the image of a model wearing the e-commerce product - $\mathbf{N}$, you have to write a concise caption for the specific product. But think step by step. First, carefully observe the $\mathbf{N}$ in the image and identify all the fine-grained visual attributes of the specific product. Then, using the identified attributes, write a concise attribute-aware caption for the specific product. Only output the final caption.
Assistant: $\tilde{\mathbf{C}}$

---

**Prompt used under explicit CoT paradigm**

<image($\mathbf{I}$)>
Given the image of a model wearing the e-commerce product - $\mathbf{N}$, you have to write a concise caption for the specific product. But think step by step. First, carefully observe the $\mathbf{N}$ in the image and list all the fine-grained visual attributes of the specific product. Then, using the identified attributes, write a concise attribute-aware caption for the specific product. Output both the attributes and the final caption strictly in json.
Assistant: $\tilde{\mathbf{C}}$

---

**Prompt used under the ICL paradigm**

<image($\mathbf{I}$)>
Given the image of a model wearing the e-commerce product - $\mathbf{N}$, write a concise caption for the specific product, incorporating its fine-grained attributes. Take reference from the provided examples.
Example1: <image ($I_1^c$)> Product name: $N_1$, Caption: $C_1$
Example2: <image ($I_2^c$)>Product name: $N_2$, Caption: $C_2$
Output only the final caption.
Assistant: $\tilde{\mathbf{C}}$

---

**Prompt used under the ICL+CoT-e paradigm**

<image($\mathbf{I}$)>
Given the image of a model wearing the e-commerce product - $\mathbf{N}$, write a concise caption for the specific product, incorporating its fine-grained attributes. But think step by step. First, carefully observe the $\mathbf{N}$ in the image and list all the fine-grained visual attributes of the specific product. Then, using the identified attributes, write a concise attribute-aware caption for the specific product. Take reference from the provided examples.
Example1: <image ($I_1^c$)> Product name: $N_1$, Caption: $C_1$
Example2: <image ($I_2^c$)>Product name: $N_2$, Caption: $C_2$
Output both the attributes and the final caption strictly in json.
Assistant: $\tilde{\mathbf{C}}$

Under the ICL paradigm, all baselines except TinyLLaVA-3B Zhou et al. (2024) are fed with two in-context samples, retrieved randomly from the training set. TinyLLaVA cannot accept multi-image inputs, hence we only feed the product name and captions of the retrieved in-context samples in TinyLLaVA.

## 4.4 Metrics

### 4.4.1 Traditional metrics

To measure the captioning performance of these VLMs, we utilize standard image-captioning metrics such as BLEU Papineni et al. (2002), ROUGE Lin (2004) and METEOR Banerjee & Lavie (2005), where higher values for all the scores are desired.

### 4.4.2 LLM-as-Judge

While the above metrics provide a general sense of fluency and lexical overlap with references, they often fail to capture whether the caption accurately describes the most relevant visual attributes, particularly in the fashion domain where small attribute errors can significantly mislead. To address this gap, we introduce a task-specific LLM-based evaluation protocol inspired by human verification, which measures how faithfully a caption covers gold-standard attribute-value pairs of a fashion product. We use LLaMA-3.2-V-11B model as a judgment engine to determine whether each attribute-value pair of the oracle data (described in section 4.7) is correctly captured in the caption. Each evaluation is phrased as a binary Yes/No question. Below is the prompt template used for this metric calculation:

> **Prompt used for LLM judge**
>
> Caption: $\{caption(\mathbf{C})\}$
> Attribute: $\{attribute(a)\}$
> USER: You are given a generated caption, and the ground-truth attributes of an e-commerce fashion product. Does the caption correctly capture the value of the above attribute? Answer 'Yes' or 'No'.
> ASSISTANT: {Yes}.

We then count the occurence of 'Yes' and get the average score for number of attributes covered per sample.

## 4.5 Implementation Details

We use PyTorch Paszke et al. (2019) and the HuggingFace Transformers library Wolf et al. (2020) for majority of our implementation. For retrieval over the product knowledge base (ProductKB) (Section 3.2.1), we employ FAISS Douze et al. (2025) for efficient nearest-neighbor search. Visual embeddings for all ProductKB entries are pre-computed using a CLIP vision encoder Radford et al. (2021). Unless otherwise specified, we use $K{=}1$ for Chain-of-Attributes and $K_{icl}{=}2$ for ICL exemplars in RA-CoA throughout the paper. For the VLMs used in our experiments, we rely on the authors' official code repositories or publicly available HuggingFace implementations, prioritizing reproducibility. All VLMs are used in a frozen, inference-only setting without any fine-tuning. Unless stated otherwise, all ablation studies are conducted using InternVL2-8B as the underlying VLM. All experiments are conducted on a single machine equipped with three NVIDIA A6000 GPUs (48GB memory each).

## 4.6 Results and Discussion

Table 1 presents quantitative comparison of our method RA-CoA with other training-free paradigms across varied-scale VLMs. RA-CoA consistently outperforms all baselines, including proprietary GPT-4o, validating its effectiveness and robustness for training-free fashion image captioning. Zero-shot methods yield poor METEOR scores (10.4–13.8), often producing generic descriptions that miss key fashion attributes (see Figure 1). Chain-of-Thought reasoning shows limited or even degraded performance, likely due to models describing the entire image rather than the product, leading to attribute omission or hallucination. In-Context Learning improves METEOR by 1.5–18 points over zero-shot, yet still struggles with complete attribute

Table 1: Comparison of RA-CoA against different prompting paradigms across VLMs of varying scales. Δ (gray rows) represents the gain of RA-CoA over Zero-shot.

| Model (# params) | Paradigm | BLEU-1 | Avg. BLEU | Rouge-1 | Rouge-2 | Rouge-L | METEOR | LLM-Judge |
|---|---|---|---|---|---|---|---|---|
| TinyLLaVA (3B) | Zero-shot | 12.3 | 3.5 | 17.1 | 2.4 | 11.4 | 12.2 | 17.2 |
| | CoT-i | 11.0 | 3.3 | 18.6 | 2.7 | 12.6 | 11.0 | 10.1 |
| | CoT-e | 7.8 | 2.3 | 12.8 | 1.8 | 8.7 | 8.6 | 8.1 |
| | ICL | 13.1 | 4.8 | 24.5 | 4.9 | 18.7 | 13.7 | 36.5 |
| | ICL+CoT-e | 2.9 | 1.2 | 9.2 | 2.3 | 7.5 | 7.9 | 15.2 |
| | **RA-CoA** | **22.1** | **11.50** | **32.5** | **10.8** | **25.2** | **24.6** | **50.9** |
| | Δ | +9.8 | +8.0 | +15.4 | +8.4 | +13.8 | +12.4 | +33.7 |
| Qwen-2.5VL (3B) | Zero-shot | 8.0 | 2.2 | 19.4 | 2.8 | 12.7 | 10.4 | 31.7 |
| | CoT-i | 4.2 | 1.2 | 22.8 | 3.4 | 14.9 | 10.0 | 29.3 |
| | CoT-e | 9.1 | 2.7 | 20.8 | 3.4 | 13.3 | 10.4 | 20.9 |
| | ICL | 26.2 | 13.5 | 29.7 | 9.7 | 23.3 | 26.5 | 49.8 |
| | ICL+CoT-e | 9.0 | 3.1 | 20.5 | 3.9 | 15.0 | 16.3 | 19.0 |
| | **RA-CoA** | **26.8** | **14.5** | **35.3** | **11.7** | **28.3** | **28.6** | **55.2** |
| | Δ | +18.8 | +12.3 | +15.9 | +8.9 | +15.6 | +18.2 | +23.5 |
| Qwen-2.5VL (7B) | Zero-shot | 8.2 | 2.2 | 18.7 | 2.6 | 11.6 | 10.9 | 33.2 |
| | CoT-i | 6.4 | 1.7 | 19.0 | 2.9 | 12.1 | 10.2 | 32.5 |
| | CoT-e | 7.5 | 2.1 | 20.4 | 3.4 | 12.9 | 10.4 | 23.0 |
| | ICL | 14.5 | 8.2 | 29.3 | 8.1 | 20.8 | 19.2 | 50.0 |
| | ICL+CoT-e | 10.6 | 4.0 | 23.1 | 5.4 | 17.6 | 16.5 | 19.8 |
| | **RA-CoA** | **31.4** | **17.7** | **39.2** | **14.2** | **32.3** | **32.4** | **57.7** |
| | Δ | +23.2 | +15.5 | +20.5 | +11.6 | +20.7 | +21.5 | +24.5 |
| InternVL2 (8B) | Zero-shot | 11.3 | 3.2 | 16.2 | 2.3 | 11.0 | 13.8 | 25.9 |
| | CoT-i | 13.7 | 4.0 | 16.6 | 3.0 | 12.2 | 15.7 | 38.4 |
| | CoT-e | 13.9 | 4.2 | 20.8 | 3.1 | 13.9 | 13.5 | 26.1 |
| | ICL | 30.2 | 17.5 | 35.2 | 13.4 | 28.7 | 31.8 | 58.9 |
| | ICL+CoT-e | 14.2 | 7.1 | 23.4 | 7.5 | 19.0 | 24.8 | 29.0 |
| | **RA-CoA** | **38.6** | **23.7** | **41.0** | **17.4** | **36.0** | **38.1** | **61.1** |
| | Δ | +27.3 | +20.5 | +24.8 | +15.1 | +25.0 | +24.3 | +35.2 |
| GPT-4o | Zero-shot | 10.6 | 2.9 | 17.7 | 2.6 | 11.4 | 12.6 | 34.2 |
| | CoT-e | 22.8 | 14.3 | 30.2 | 12.8 | 23.9 | 25.4 | 37.5 |
| | ICL | 46.6 | 32.1 | 53.8 | 28.6 | 46.3 | 50.3 | 60.7 |
| | **RA-CoA** | **63.2** | **48.8** | **69.3** | **44.2** | **62.9** | **67.6** | **76.5** |
| | Δ | +52.6 | +45.9 | +51.6 | +41.6 | +51.5 | +55.0 | +42.3 |

coverage. Chain-of-thought reasoning combined with in-context learning underperforms ICL alone, indicating that these models struggle to follow unstructured explicit reasoning instructions reliably. Adding ICL exemplars partially recovers this degradation compared to CoT-e, but the inherent weakness of unstructured explicit reasoning still limits overall performance. In contrast, RA-CoA achieves the highest performance across all models. For TinyLLaVA (3B), it doubles the METEOR score (24.6 vs. 12.2), improving 79% over ICL. Gains scale with model size. RA-CoA improves METEOR by 12.4 points for TinyLLaVA (3B) and 24.3 for InternVL2 (8B) over zero-shot. Larger models perform better overall, with InternVL2 (8B) achieving the best open-source results (METEOR: 38.1, LLM Judge: 61.1). GPT-4o, when used with RA-CoA, achieves the highest scores (METEOR: 67.6, LLM Judge: 63.5), though its zero-shot performance is on par with smaller open-source models. These findings affirm that RA-CoA's structured, attribute-focused approach enables more accurate and interpretable captions, with gains increasing alongside model scale.

Qualitative examples in Figure 3 further illustrate RA-CoA's superior caption fidelity across diverse fashion categories. The generated descriptions consistently capture fine-grained product attributes with high accuracy across different product categories.

## 4.7 Ablations and Analysis

We conduct the following ablation studies to validate the design choices and quantify the impact of different components of our method RA-CoA:

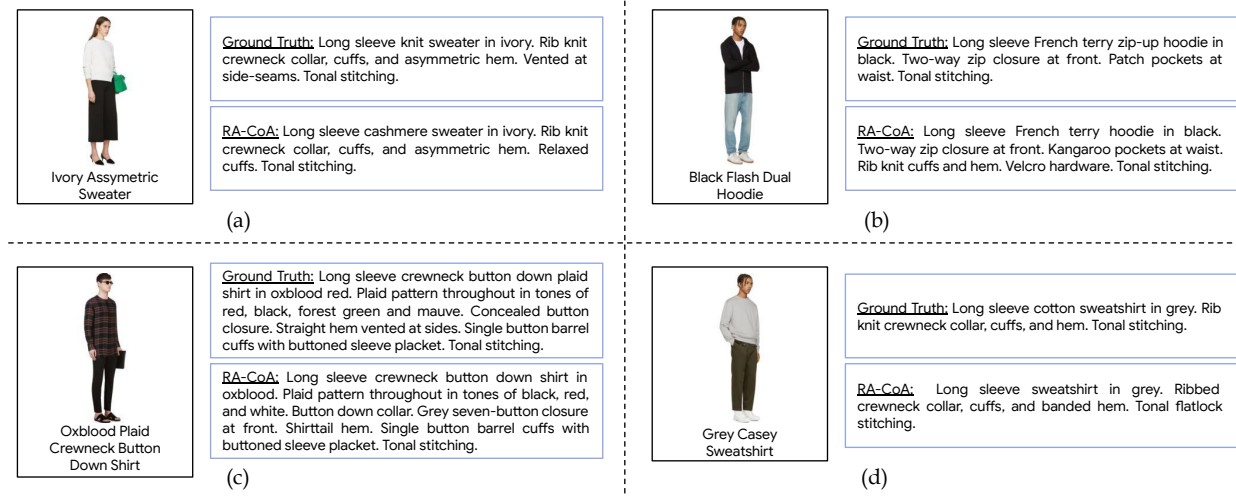

Figure 3: A selection of RA-CoA's results. Our proposed approach RA-CoA generates coherent captions with minimal hallucinations that align well with the human-annotated ground truth descriptions.

Table 2: Ablation study to quantify the impact of top-K retrievals for CoA in our proposed RA-CoA method.

| Model (# params) | Top-K | BLEU-1 | Avg. BLEU | Rouge-1 | Rouge-2 | Rouge-L | METEOR |
|---|---|---|---|---|---|---|---|
| TinyLLaVA (3B) | 1 | 22.1 | 11.50 | 32.5 | 10.8 | 25.2 | 24.6 |
| | 2 | 20.0 | 9.8 | 30.5 | 9.4 | 23.2 | 22.1 |
| | 3 | 18.7 | 8.9 | 29.2 | 8.6 | 21.9 | 21.0 |
| Qwen-2.5VL (3B) | 1 | 26.8 | 14.5 | 35.3 | 11.7 | 28.3 | 28.6 |
| | 2 | 31.3 | 16.6 | 37.7 | 12.6 | 29.3 | 31.3 |
| | 3 | 25.6 | 13.1 | 33.1 | 9.8 | 25.1 | 28.3 |
| Qwen-2.5VL (7B) | 1 | 31.4 | 17.7 | 39.2 | 14.2 | 32.3 | 32.4 |
| | 2 | 31.3 | 16.6 | 37.7 | 12.6 | 29.4 | 31.3 |
| | 3 | 29.8 | 15.4 | 36.6 | 11.7 | 27.7 | 30.8 |
| InternVL2 (8B) | 1 | 38.6 | 23.7 | 41.0 | 17.4 | 36.0 | 38.1 |
| | 2 | 38.3 | 22.9 | 40.2 | 15.8 | 33.3 | 38.3 |
| | 3 | 36.5 | 21.6 | 38.8 | 14.9 | 31.6 | 37.5 |

### 4.7.1 Optimal top-K

In RA-CoA, attributes are inferred from retrieved products (Section 3.2.1) and subsequently used to guide caption generation (Section 3.2.4). A key design choice in this process is the number of retrieved samples (top-K) used for attribute aggregation (Eq. 1). While increasing K can potentially improve attribute coverage by incorporating more diverse evidence, it may also introduce weakly aligned or irrelevant attributes, particularly in fine-grained fashion domains.

To study this trade-off, we vary K in {1, 2, 3} and report the results in Table 2. We observe that captioning performance generally decreases as K increases. This trend is consistent with the increased presence of noise and redundancy at higher K values, which can dilute the relevance of the aggregated attribute set. In addition, conditioning on larger attribute sets may increase the tendency of VLMs to include unsupported details, affecting caption precision. Overall, these results suggest that a compact set of high-confidence attributes obtained from the single closest retrieved sample (K=1) provides a more favorable balance between attribute relevance and captioning accuracy for most models.

Table 3: Ablation study to (1) quantify the importance of retrieval-augmented ICL exemplars, and (2) quantify the gap of RA-CoA with respect to oracle variants.

| Model (# params) | Method variant | BLEU-1 | Avg.BLEU | Rouge-1 | Rouge-2 | Rouge-L | METEOR |
|---|---|---|---|---|---|---|---|
| TinyLLaVA (3B) | RA-CoA | 22.1 | 11.50 | 32.5 | 10.8 | 25.2 | 24.6 |
| | RA-CoA w/o ICL exemplars | 16.8 | 5.3 | 19.8 | 2.5 | 12.8 | 17.9 |
| | RA-CoA with Oracle Attributes | 22.9 | 11.2 | 34.1 | 10.5 | 26.4 | 24.7 |
| | RA-CoA with Oracle Attribute-Values | 53.9 | 44.9 | 74.0 | 60.4 | 71.3 | 71.6 |
| Qwen-2.5VL (3B) | RA-CoA | 26.8 | 14.5 | 35.3 | 11.7 | 28.3 | 28.6 |
| | RA-CoA w/o ICL exemplars | 17.7 | 5.7 | 22.6 | 2.8 | 14.2 | 19.8 |
| | RA-CoA with Oracle Attributes | 30.6 | 16.4 | 40.2 | 12.6 | 31.8 | 31.8 |
| | RA-CoA with Oracle Attribute-Values | 69.4 | 62.8 | 87.4 | 77.2 | 83.8 | 82.4 |
| Qwen-2.5VL (7B) | RA-CoA | 31.4 | 17.7 | 39.2 | 14.2 | 32.3 | 32.4 |
| | RA-CoA w/o ICL exemplars | 17.9 | 5.7 | 23.8 | 2.8 | 14.3 | 21.7 |
| | RA-CoA with Oracle Attributes | 33.4 | 18.1 | 44.0 | 14.6 | 35.6 | 34.1 |
| | RA-CoA with Oracle Attribute-Values | 86.5 | 83.3 | 94.6 | 91.0 | 93.4 | 91.7 |
| InternVL2 (8B) | RA-CoA | 38.6 | 23.7 | 41.0 | 17.4 | 36.0 | 38.1 |
| | RA-CoA w/o ICL exemplars | 12.8 | 4.1 | 22.7 | 2.8 | 14.6 | 19.8 |
| | RA-CoA with Oracle Attributes | 45.1 | 27.0 | 47.2 | 18.7 | 40.8 | 42.5 |
| | RA-CoA with Oracle Attribute-Values | 95.7 | 93.8 | 98.2 | 95.5 | 97.4 | 97.1 |

### 4.7.2  Contribution of ICL exemplars in RA-CoA

As elaborated in Section 3.2.4, RA-CoA leverages retrieved products as in-context exemplars during caption generation, to guide the VLM for structure and style of the generated caption. In this ablation, to understand the impact of in-context exemplars in the quality of caption generation, we evaluate RA-CoA without exemplars during the caption generation. Table 3 presents the quantitative results. Even without exemplars, RA-CoA consistently outperforms zero-shot, CoT-i, and CoT-e approaches, indicating the effectiveness of RA-CoA's attribute-centric reasoning. At the same time, removing exemplars leads to a noticeable but expected performance drop compared to the full RA-CoA. For example, METEOR decreases from 26.8 to 17.7 for Qwen2VL-3B and from 38.1 to 19.8 for InternVL2-8B. This demonstrates the critical role of retrieved exemplars in generating coherent, stylistically appropriate captions that effectively incorporate the identified attributes.

### 4.7.3  Comparison with Oracle Variants

To further study the contribution of different components in RA-CoA, we implement two oracle variants. Since original ground-truth attribute-value pairs are not available in FashionGen dataset, we obtain these pairs from expert-written captions similar to our ProductKB construction (Section 3.1) and consider these extracted pairs as ground truth for evaluation purposes.

**Caption Generation with Oracle Attributes.** Our RA-CoA method relies on retrieval to identify relevant attributes for a fashion item. To understand how close our retrieval mechanism comes to optimal attribute identification, we implement an Attribute-Oracle variant that directly uses ground truth attribute keys. Table 3 reveals that while Attribute-Oracle consistently outperforms standard RA-CoA, the gains are relatively modest across most models. TinyLLaVA (3B) improves by just 0.1 METEOR points (24.7 vs 24.6). The gap widens somewhat for larger models, with InternVL2 (8B) showing a 4.4-point improvement (42.5 vs 38.1). These results are encouraging, suggesting our retrieval-based attribute identification approaches the theoretical ceiling, particularly for smaller models. The increased gap for larger models indicates that as VLM capability grows, the limiting factor shifts more toward attribute identification quality rather than the model's ability to leverage those attributes.

**Caption Generation with Oracle Attribute-Values.** To establish the theoretical upper limit of our approach, we implement an Attribute-value-pair-Oracle variant that directly uses gold-standard attribute-value pairs extracted from expert captions. Table 3 shows dramatic performance improvements across all models. TinyLLaVA (3B) reaches 71.6 METEOR (vs 24.6 for standard RA-CoA), while InternVL2 (8B) achieves 97.1 METEOR (vs 38.1). Unlike the modest gains from Attribute-Oracle, these substantial improvements reveal that while RA-CoA effectively identifies relevant attributes, the primary performance bottleneck

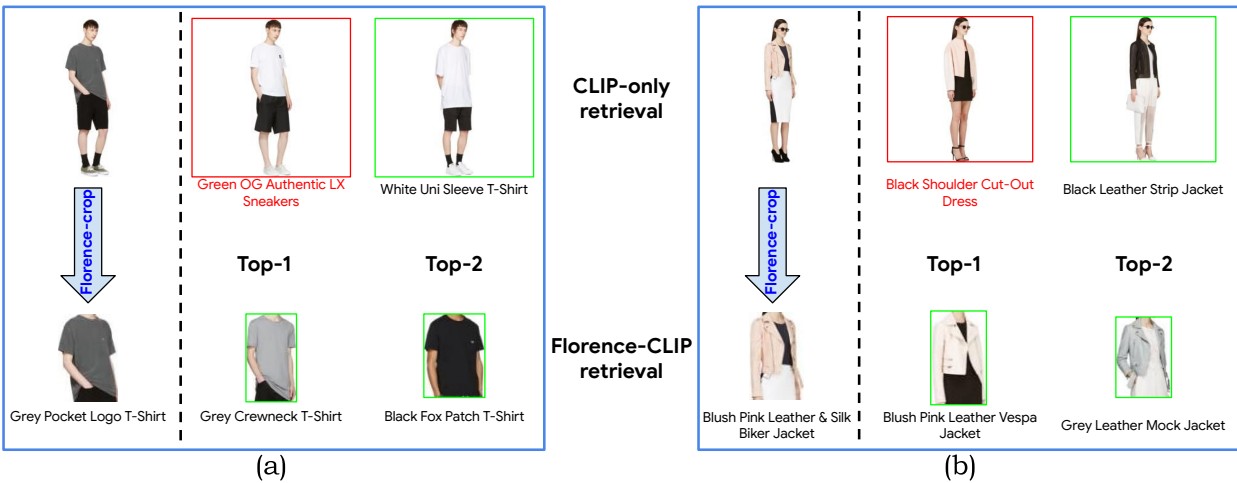

Figure 4: Comparison of CLIP-only and Florence-CLIP for retrieval in RA-CoA (Section 3.2.1).

Table 4: Comparison of using full image vs. Florence-2–cropped product region as VLM input for captioning. Cropping offers a modest gain while the full image already yields strong results. All results are reported for InternVL2 (8B) as base VLM.

| Paradigm | Input image | BLEU-1 | Avg.BLEU | Rouge-1 | Rouge-2 | Rouge-L | METEOR |
|---|---|---|---|---|---|---|---|
| Zeroshot | | 9.6 | 2.7 | 19.1 | 2.9 | 12.6 | 11.7 |
| CoT-i | | 13.0 | 3.8 | 18.0 | 2.9 | 12.6 | 15.0 |
| CoT-e | Cropped Product | 7.0 | 2.1 | 16.8 | 2.4 | 12.4 | 13.2 |
| ICL | | 29.8 | 17.3 | 35 | 13.3 | 28.7 | 31.6 |
| RA-CoA | | **39.4** | **24.3** | **41.8** | **18.0** | **36.7** | **38.8** |
| RA-CoA | Full Image | 38.6 | 23.7 | 41.0 | 17.4 | 36.0 | 38.1 |

lies in attribute value generation. VLMs struggle significantly with accurately determining attribute values from visual input, despite correctly identifying which attributes to consider. The widening gap with larger models suggests that as VLM capability increases, the limiting factor becomes increasingly centered on value prediction accuracy, pointing to a clear direction for future improvements in the CoA mechanism.

### 4.7.4 Effect of product-aware cropping on retrieval and captioning

To isolate the impact of our retrieval backbone and examine the role of product-aware cropping in both retrieval and captioning, we perform a two-part analysis.

**Retrieval.** We compare two setups: (1) *Global CLIP-only Retrieval* Radford et al. (2021), which retrieves similar samples using embeddings of the full image instead of cropped ones, and (2) *Florence–CLIP Retrieval*, which first detects and crops the target product region using Florence-2 Xiao et al. (2024)(as described in Section 3.1) before embedding it with CLIP. Figure 4 illustrates representative retrieval errors from the CLIP-only approach, where visually unrelated items (e.g., "*Green OG Authentic LX Sneakers*" retrieved for a "*Grey Pocket Logo T-Shirt*") are selected due to background similarity. In contrast, the Florence–CLIP variant consistently retrieves visually and semantically aligned products by focusing on the correct region of interest. This refinement substantially reduces noisy retrievals and establishes a cleaner foundation for downstream attribute aggregation and caption generation.

**Captioning.** While product-aware cropping is essential for retrieval, during caption generation we input the full image to the Vision-Language Model (VLM). This choice is motivated by the fact that modern instruction-tuned VLMs exhibit strong coarse-level grounding abilities and can attend to the relevant product when guided by its name in the prompt. To further validate this design, we conducted an experiment where

Table 5: Effect of ProductKB size on captioning performance using InternVL2-8B within the RA-CoA framework. Performance improves with KB size, but gains saturate beyond 20K entries, indicating that RA-CoA remains data-efficient even with smaller knowledge bases.

| ProductKB size | BLEU-1 | Avg.BLEU | ROUGE-1 | ROUGE-2 | ROUGE-L | METEOR |
|---|---|---|---|---|---|---|
| 10K | 36.2 | 21.1 | 38.8 | 15.1 | 33.4 | 35.5 |
| 20K | 37.6 | 22.7 | 40.3 | 16.6 | 35.2 | 37.1 |
| 40K | 38.5 | 23.5 | 41.2 | 17.2 | 35.9 | 37.9 |
| 60K (Full) | **38.1** | **23.7** | **41.0** | **17.4** | **36.0** | **38.1** |

Table 6: Effect of noisy ProductKB entries on captioning performance using InternVL2-8B within the RA-CoA framework. Performance decreases with increasing noise in attribute values, suggesting that RA-CoA is relatively robust to corrupted ProductKB entries.

| % corrupted | BLEU-1 | Avg.BLEU | ROUGE-1 | ROUGE-2 | ROUGE-L | METEOR |
|---|---|---|---|---|---|---|
| 0% | **38.1** | **23.7** | **41.0** | **17.4** | **36.0** | **38.1** |
| 25% | 37.0 | 21.8 | 39.5 | 16.1 | 34.5 | 36.5 |
| 50% | 36.9 | 21.3 | 39.2 | 15.3 | 33.9 | 35.9 |
| 75% | 35.7 | 19.9 | 37.7 | 13.8 | 32.4 | 34.3 |

the VLM (InternVL2-8B) received the Florence-2–cropped product region instead of the full image. We observed a modest METEOR gain of +0.8 in RA-CoA (38.9 vs. 38.1), suggesting that while explicit cropping can offer a small improvement, the model already performs effectively with full images. Hence, cropping is *necessary for retrieval* but only *optional for caption generation*, providing marginal benefit when used. Results are summarized in Table 4.

### 4.7.5 Effect of ProductKB Quality and Scale

Since RA-CoA relies on a structured Product Knowledge Base (ProductKB) for retrieval-augmented reasoning, its performance may depend on the quality, completeness, and scale of this knowledge source. To examine this, we conduct a series of controlled experiments analyzing three aspects: (i) varying KB size, (ii) introducing noise into KB entries (Noisy KB), and (iii) simulating sparsity in attribute annotations (Sparse KB).

**Varying Knowledge Base Size.** We progressively subsample the ProductKB to contain 10K, 20K, 40K, and 60K entries, and evaluate RA-CoA using InternVL2-8B as the VLM. Table 5 summarizes the results. The corresponding METEOR scores are 36.9, 35.5, 37.1, 37.9, and 38.1, respectively. While performance improves with KB size, gains beyond 20K are incremental. This suggests that even moderately sized KBs capture sufficient diversity for most fashion products, and larger KBs mainly help when new or niche SKUs are introduced. Average metrics over the full test set may not fully reflect improvements on such long-tail items, where retrieval quality depends directly on KB diversity and scale.

**Noisy Knowledge Base.** To evaluate robustness to noisy entries in the ProductKB, we introduce noise by shuffling attribute values in 25%, 50%, and 75% of the KB entries (e.g., replacing half-sleeve with full-sleeve or velcro with zip). Table 6 summarizes the results. As the noise level increases, METEOR scores decrease slightly from 38.1 (original KB) to 36.5, 35.9, and 34.3, respectively. This limited degradation highlights the robustness of RA-CoA, which stems from a key design choice: post retrieval during *Chain-of-Attributes*, we use only the attribute *keys* from retrieved samples, not their values. This prevents noisy attribute values from being propagated while still guiding the model on which attributes to predict.

**Sparse Knowledge Base.** To analyze the effect of incomplete attribute annotations, we simulate a sparse ProductKB by randomly dropping 25%, 50%, and 75% of key–value pairs from each KB entry. Results are summarized in Table 7. As sparsity increases, METEOR scores decline from 38.1 to 33.9, 29.7, and 25.5, respectively. While moderate sparsity (25%) has limited impact, performance degrades substantially at higher

Table 7: Effect of sparse ProductKB entries on captioning performance using InternVL2-8B within the RA-CoA framework. Performance degrades as sparsity in the ProductKB increases, as it affects retrieval quality and attribute guidance.

| % key-value pairs dropped | BLEU-1 | Avg.BLEU | ROUGE-1 | ROUGE-2 | ROUGE-L | METEOR |
|:---:|:---:|:---:|:---:|:---:|:---:|:---:|
| 0% | **38.1** | **23.7** | **41.0** | **17.4** | **36.0** | **38.1** |
| 25% | 34.4 | 20.5 | 39.2 | 16.4 | 34.3 | 33.9 |
| 50% | 28.4 | 17.1 | 37.6 | 16.4 | 33.1 | 29.7 |
| 75% | 21.1 | 13.1 | 35.9 | 16.7 | 31.6 | 25.5 |

Table 8: Quantitative comparison of RA-CoA with UniFashion Zhao et al. (2024) (prior SOTA supervised method) showcasing real-world generalization of RA-CoA despite being completely training-free.

| Method | Fine-tuning | BLEU-1 | Avg.BLEU | Rouge-1 | Rouge-2 | ROUGE-L | METEOR |
|:---|:---:|:---:|:---:|:---:|:---:|:---:|:---:|
| UniFashion | ✓ | 11.4 | 3.9 | 16.5 | 4.1 | 12.5 | 16.2 |
| **RA-CoA (Ours)** | ✗ | **22.0** | **7.7** | **26.1** | **5.7** | **18.3** | **20.2** |

sparsity levels (75%). This degradation can be attributed to two factors. First, fewer available attribute keys reduce explicit guidance on which visual properties the model should reason about. Second, and more critically, sparse KB entries weaken retrieval quality, as retrieved products become less semantically aligned with the query. Since these retrieved samples are also used as in-context exemplars during caption generation, reduced alignment corrupts exemplar quality and leads to weaker captions. Together, these effects highlight the importance of sufficiently informative KB entries for both attribute-centric reasoning and exemplar-guided caption synthesis in RA-CoA.

#### 4.7.6 Comparison with prior SOTA and Generalization of RA-CoA beyond FashionGen dataset

Table 8 shows quantitative comparison of RA-CoA with prior state-of-the-art approach, UniFashion Zhao et al. (2024). Notably, UniFashion is a fully supervised method, explicitly trained on large fashion datasets including FashionGen. Thus, direct evaluation of UniFashion on the FashionGen test set would be an unfair comparison with our training-free RA-CoA. To enable a fair and unbiased evaluation, we curated 500 fashion product images with their corresponding captions from the web[3] and evaluated both approaches on this held-out real-world set. In this setting, our training-free RA-CoA (with InternVL2-8B) consistently outperforms the supervised UniFashion (7B) baseline across all evaluation metrics, achieving gains of 10.6 points in BLEU-1, 3.8 points in average BLEU, 9.6 points in ROUGE-1, 1.6 points in ROUGE-2, 5.8 points in ROUGE-L, and 4.0 points in METEOR. The overall low scores can be attributed to the difference in style of ground truth captions and the captions in ProductKB (derived from FashionGen dataset). This comparison highlights the performance gap that arises between training-intensive and training-free paradigms under real-world distribution shift, demonstrating the robustness and scalability of RA-CoA in practical deployment scenarios.

#### 4.7.7 Generalization of RA-CoA beyond e-commerce Fashion domain

To assess the generalizability of our proposed approach beyond fashion domain, we evaluate RA-CoA on the Furniture Dataset Nowicki et al. (2025) with structured attribute annotations. Unlike the FashionGen dataset where product captions are available, this dataset provides ground truth attribute-value pairs directly; we generate natural language captions from these ground truth attribute-value pairs using Qwen3-8B LLM. The training split is used as the ProductKB and evaluation is performed on the test split. Table 9 presents quantitative comparison of RA-CoA with other VLM paradigms. RA-CoA outperforms all baselines across all metrics. Two patterns are worth noting. First, CoT achieves a strong LLM-judge score (75.1) despite lower lexical scores, suggesting that furniture attributes are visually salient enough for free-form reasoning to identify them correctly, but CoT lacks the exemplar-guided structure to produce well-formed captions.

---

[3]www.amazon.com

Table 9: Quantitative comparison of RA-CoA with other VLM paradigms on the Furniture Dataset, showcasing generalization of our approach beyond fashion domain.

| Method | BLEU-1 | Avg.BLEU | Rouge-1 | Rouge-2 | ROUGE-L | METEOR | LLM-Judge |
|---|---|---|---|---|---|---|---|
| Zero-shot | 4.0 | 1.4 | 22.8 | 2.0 | 17.4 | 9.2 | 65.7 |
| CoT-i | 38.6 | 17.7 | 36.5 | 11.2 | 28.3 | 27.4 | 75.1 |
| ICL | 71.0 | 55.0 | 68.2 | 44.8 | 67.5 | 72.8 | 34.2 |
| **RA-CoA (Ours)** | **74.5** | **60.6** | **74.8** | **51.3** | **71.4** | **73.2** | **77.2** |

Table 10: User preference study over 300 unique test images using InternVL2-8B as the backbone. RA-CoA is preferred in 68.6% of cases, reflecting higher perceived coherence, completeness, and usefulness.

| Method | Preference (%) |
|---|---|
| Zero-shot | 3.7 |
| CoT-i (Implicit CoT) | 6.7 |
| CoT-e (Explicit CoT) | 7.0 |
| In-Context Learning (ICL) | 14.0 |
| **RA-CoA (Ours)** | **68.6** |

Second, ICL achieves high lexical scores but a dramatically lower LLM-judge score (34.2), consistent with exemplar style copying that inflates n-gram overlap without improving attribute correctness. RA-CoA resolves both limitations: CoA ensures accurate attribute grounding while retrieved exemplars guide caption structure, achieving the highest scores across all metrics. These results confirm that RA-CoA's structured attribute decomposition generalizes to any domain where fine-grained attribute reasoning over a structured KB is required.

### 4.7.8 User preference study

To complement automatic evaluation, we conducted a user preference study to assess the qualitative usefulness of captions generated by different paradigms. We recruited six human evaluators, each of whom rated 50 randomly sampled, disjoint test images, resulting in 300 unique evaluation instances. For each image, users were shown anonymized captions generated by InternVL2-8B under all evaluated VLM paradigms. The order of captions was randomized to mitigate positional bias. User preferences are summarized in Table 10. Zero-shot captions are rarely preferred, highlighting the challenge of generating complete and user-aligned descriptions without additional guidance. Both implicit and explicit CoT variants provide only marginal gains over Zero-shot, suggesting that reasoning traces alone, without explicit grounding in retrieved attribute knowledge, do not substantially improve caption quality. Although ICL outperforms Zero-shot and CoT-based methods, it remains considerably less preferred than RA-CoA, indicating that exemplar-based prompting is insufficient for consistently high-quality caption generation. Overall, these results emphasize the importance of retrieval-augmented, attribute-aware reasoning for producing captions that better align with human expectations of coherence, completeness, and usefulness.

### 4.7.9 Attribute-wise sensitivity analysis

To better understand attribute-level behavior beyond holistic caption quality, we conduct an explicit attribute-wise sensitivity analysis that evaluates how accurately different attributes are captured in the generated captions. Specifically, we extract the attribute values from the generated captions using LLaMA-3.2-8B-Instruct Grattafiori et al. (2024), and compare them against ground-truth attribute labels. Exact string matching is insufficient in this setting due to frequent lexical variability among semantically equivalent attributes (e.g., crew-neck vs. round-neck, zip vs. zipper, or slip-on vs. no-closure). To address this, we adopt a hybrid evaluation protocol: attributes that exactly match the labels are assigned a score of 5, while non-exact matches are evaluated using a LLaMA-based semantic similarity score on a 0–5 scale Mañas et al. (2024).

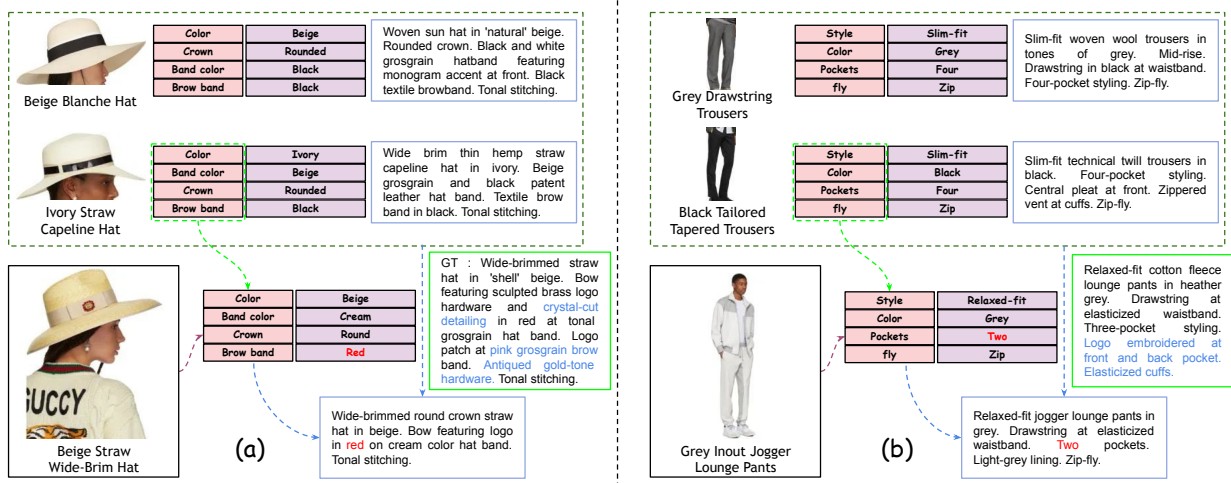

Figure 5: Error analysis into RA-CoA's generations. The text in red denotes incorrect/hallucinated features. The text in blue in the GT caption indicates features missed in the generated caption.

Table 11: Computational latency analysis of RA-CoA's components.

| Step | Time (seconds) |
|---|---|
| ROI crop using Florence-2 | 0.30 |
| CLIP embedding extraction for ROI | 0.04 |
| Retrieval from ProductKB (Section 3.2.1) | 0.01 |
| Chain-of-attributes (Section 3.2) | 1.40 |
| Attribute-value aware captioning (Section 3.2.4) | 1.80 |
| **Average inference time per sample (Total)** | **3.5** |

From this experiment, we observe the following: *(i) High-confidence or less sensitive attributes*: RA-CoA achieves higher accuracy (avg. score $> 3.5$) for the attributes that are visually prominent such as color, closure, sleeve type, front print, and hood. *(ii) Low-confidence or sensitive attributes*: RA-CoA achieves lower accuracy (average score $< 2.5$) for attributes that are often occluded, subtle, or viewpoint-sensitive, such as, hemline, fly type, and pockets.

### 4.7.10 Error analysis

We conducted an error analysis of the captions generated by RA-CoA to understand their shortcomings. While RA-CoA produces coherent captions with reduced hallucination, some limitations remain and can be broadly categorized into two types: (i) missing attributes and (ii) hallucinated values. An illustration of these error types is shown in Figure 5. These issues arise from gaps in attribute coverage during retrieval and the VLM's limited grounding between visual cues and attribute semantics. These challenges could be mitigated by (i) expanding the ProductKB to include a more diverse and comprehensive set of product samples, and (ii) performing one-time pretraining of VLMs on structured attribute–value annotations to improve semantic grounding of product attributes, which we leave for future work.

### 4.7.11 Latency analysis

To assess the practical feasibility of RA-CoA for real-world e-commerce applications, we analyze the per-sample inference time of our framework. Table 11 reports the computational breakdown averaged across 1,000 samples on a single NVIDIA A6000 GPU. The total average inference time per sample is approximately 3.5 seconds, with the primary computational bottleneck being the two-stage VLM prompting: chain-of-attribute guided value generation (1.4s) and attribute-value aware caption generation (1.8s). In contrast, the retrieval

is highly efficient. ROI extraction using Florence-2 takes 0.3s, CLIP embedding extraction requires only 0.04s, and retrieval from a ProductKB of 60K samples via Faiss indexing completes in 0.01s. This latency profile demonstrates that RA-CoA achieves practical inference speeds suitable for e-commerce applications.

## 5 Conclusion

In this work, we proposed RA-CoA, a training-free, model-agnostic framework for fashion image captioning that enhances attribute-level precision and interpretability by disentangling captioning into chain-of-attribute reasoning followed by generation. Unlike supervised methods that require frequent retraining to adapt to evolving fashion trends and vocabularies, RA-CoA leverages a product knowledge base and prompts frozen VLMs to infer relevant attributes and synthesize coherent captions. Extensive evaluations demonstrate that our approach reduces hallucination, improves caption faithfulness, and supports scalable, real-world deployment in dynamic fashion environments.

## 6 Limitations and Future Work

Despite RA-CoA's improvements in fashion image captioning, there are a few limitations: (i) Performance depends on the quality and diversity of the ProductKB. Current knowledge base and evaluations focus on western fashion vocabularies and may inadequately capture cultural-specific fashion elements. Such biases or gaps in the retrieval database may propagate to the generated captions. Future work could further validate our approach on culturally diverse sets. (ii) As revealed by our oracle experiments, VLMs struggle with inferring the values for fine-grained product attributes, particularly for subtle characteristics like design on buttons or small text on garments. Future work could explore pretraining of VLMs to improve semantic grounding of product attributes for better attribute inference.

## 7 Ethical Considerations

While RA-CoA offers accessibility benefits in fashion e-commerce, we acknowledge several ethical considerations. FashionGen contains human models wearing fashion products, raising privacy concerns and potential demographic biases. We mitigate these by using the dataset under its original license, and avoiding identity analysis. Our retrieval mechanism may still propagate biases if the ProductKB lacks diversity in represented styles or cultural demographics. The approach could theoretically be misused for misleading product descriptions, though our attribute-grounded design provides inherent safeguards. Future work should explore dataset de-identification, demographic fairness measures, and further reduction of computational requirements while improving attribute inference capabilities.

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
