# OpenReview forum: "RA-CoA: Training-free Fashion Image Captioning via Retrieval-Augmented Chain-of-Attributes"
_TMLR — Accepted by TMLR_

### Review · Reviewer_1k3U · 2026-03-15

**Summary Of Contributions:**

The paper presents the Retrieval-Augmented Chain-of-Attributes (RA-CoA) method for fashion image caption generation. The main contribution of this work includes
(1) A training-free approach that combines ideas of retrieval-augmented generation and chain-of-thought reasoning with attribute identification of fashion images.
(2) Superior results reflected by extensive comparisons with baselines including zero-shot, in-context learning, implicit and explicit chain-of-thought prompting methods.
(3) Promising generalization capability on Internet-based data.

**Audience:**

Yes

**Audience Explanation:**

The paper proposes a straightforward combination of existing techniques for the fashion image caption generation task. From the machine learning perspective, the paper is not that interesting. It may attract readers working on this specific task.

**Broader Impact Concerns:**

The paper discussed some ethical considerations regarding privacy and their solutions, which sounds OK to me.

**Claims And Evidence:**

No

**Claims Explanation:**

The claimed contributions are generally well-supported. However, I do have some concerns/questions regarding the experimental settings.

First, while the adopted baselines are reasonable, I wonder how it would perform by combining some into a stronger one. For example, combining few-shot prompting and explicit chain-of-thought seems more similar to the proposed method.

Second, it seems there are some setting discrepancies. While the method uses Florence-2, the baselines do not. In addition, the prompts used for ICL paradigm and the method are not the same: while the prompt used for caption generation emphasizes "in-context examples for style and structure", the prompt for ICL does not include such hints. The paper tests and uses K=1, while methods for comparisons still seem to use K=2. Would these differences lead to unfair comparisons?


Third, the generalization experiment is conducted on a dataset with merely 50 images. How could this small-scale dataset represent the diversity of real-world data?

Justifying the experimental settings and providing convincing results regarding the above three points are essential to verify the second and third contributions.

**Requested Changes:**

To verify the claimed contributions, the paper needs to justify the experimental settings to ensure a fair comparison, provide more convincing comparisons with stronger baselines, and more convincing results to verify the generalization claim.

In addition, there are several method designs that have not been well verified.

(1) Noisy Knowledge Base paragraph, how does this experiment of adding noise value verify the choice of using attributes instead of values? Where are the results of using attribute only and using attributes+values?

(2) While including a user study is appreciated, the paper needs to convince others that the selected six evaluators represent diverse users.

(3) According to Table 5, the results of 40K are very close (better in two metrics, while lower in four with 0.1 or 0.2) to those of 60K. The paper needs to explain why an additional 20K is necessary.

(4) Caption Generation with Oracle Attributes paragraph, I am not sure where to obtain the ground truth attribute keys.

To strengthen the relevance to TMLR's readership, the paper needs to highlight its insights behind a simple combination of RAG and CoT. Fine-grained attributes could be one point, yet whether this method can generalize to other tasks that require understanding fine-grained attributes is currently unknown.

---

> ### Author Response · Authors · 2026-04-21
> **Author response to Reviewer 1k3U**
>
> We thank the reviewer for their thoughtful review and feedback. We address the reviewer’s concerns below.
>
> **Comment-1**: First, while the adopted baselines are reasonable, I wonder how it would perform by combining some into a stronger one. For example, combining few-shot prompting and explicit chain-of-thought seems more similar to the proposed method.
>
> **Resp.**: Thank you for this suggestion. Indeed combining few-shot prompting (ICL) and explicit chain-of-thought (CoT-e) constitutes a stronger baseline, and we have now added this experiment. The results are summarized in the table below.
>
> | Model | | BLEU-1 | Avg. BLEU | ROUGE-1 | ROUGE-2 | ROUGE-L | METEOR |
> |---|---|---|---|---|---|---|---|
> | Qwen-2.5VL (3B) | ICL+CoT-e | 9.0 | 3.1 | 20.5 | 3.9 | 15.0 | 16.3 |
> | | **Ours** | **26.8** | **14.5** | **35.3** | **11.7** | **28.3** | **28.6** |
> | Qwen-2.5VL (7B) | ICL+CoT-e | 10.6 | 4.0 | 23.1 | 5.4 | 17.6 | 16.5 |
> | | **Ours** | **31.4** | **17.7** | **39.2** | **14.2** | **32.3** | **32.4** |
> | InternVL2-8B | ICL+CoT-e | 14.2 | 7.1 | 23.4 | 7.5 | 19.0 | 24.8 |
> | | **Ours** | **38.6** | **23.7** | **41.0** | **17.4** | **36.0** | **38.1** |
>
> We observe that RA-CoA consistently outperforms ICL+CoT-e across all models and metrics. For instance, on InternVL2-8B, RA-CoA achieves a METEOR score of 38.1 against 24.8 for ICL+CoT-e, a gain of 13.3 points. We also note that ICL+CoT-e underperforms ICL alone, which is consistent with a trend already visible in Table 1: CoT-e underperforms even zero-shot for most open-source models, indicating that these models struggle to follow unstructured explicit reasoning instructions reliably. Adding ICL exemplars partially recovers this degradation, but the inherent weakness of unstructured explicit reasoning still limits overall performance. This further reinforces our core motivation: RA-CoA replaces free-form reasoning with focused, one-attribute-at-a-time queries via Chain-of-Attributes, providing the structured visual grounding that ICL+CoT-e lacks. **We have added this result in Table-1 in the revised manuscript.**
>
> **Comment-2**: While the method uses Florence-2, the baselines do not.
>
> **Resp**: We would like to clarify that our method RA-CoA utilizes Florence-2 cropped images during Chain-of-Attributes (Section 3.2.3) and in the construction of ProductKB, not during product captioning (Fig. 2B). In Table 4, we performed an additional evaluation on how utilizing the cropped image for captioning as well impacts the performance.
>
> Based on the reviewer feedback, we now evaluate all baselines with Florence-2 cropped images as input (for captioning) (with InternVL2-8B model). The results are summarized below. (The last two rows are from Table 4).
>
> | Baseline | BLEU-1 | Avg. BLEU | ROUGE-1 | ROUGE-2 | ROUGE-L | METEOR |
> |---|---|---|---|---|---|---|
> | Zeroshot | 9.6 | 2.7 | 19.1 | 2.9 | 12.6 | 11.7 |
> | CoT-i | 13.0 | 3.8 | 18.0 | 2.9 | 12.6 | 15.0 |
> | CoT-e | 7.0 | 2.1 | 16.8 | 2.4 | 12.4 | 13.2 |
> | ICL | 29.8 | 17.3 | 35.0 | 13.3 | 28.7 | 31.6 |
> | **Ours (full image)** | **38.6** | **23.7** | **41.0** | **17.4** | **36.0** | **38.1** |
> | **Ours (Florence-2 cropped image)** | **39.4** | **24.3** | **41.8** | **18.0** | **36.7** | **38.8** |
>
> RA-CoA continues to outperform all baselines by a significant margin, confirming that use of cropped images in baselines for captioning does not influence their performance significantly. **We have added these results in Table 4 of the manuscript.**
>
> **Comment-3**: In addition, the prompts used for ICL paradigm and the method are not the same: while the prompt used for caption generation emphasizes "in-context examples for style and structure", the prompt for ICL does not include such hints.
>
> **Resp.**: Thank you for pointing this out. This was indeed a typo in the manuscript. The phrase "for style and structure" was part of an earlier version of the prompt that was subsequently revised during our experiments. The final prompts used for both ICL and RACoA caption generation do not include such hints and the reported results are fair and consistent. **We rectified this in the updated manuscript.**
>
> Further, we also report ICL with the style-and-structure hint added to the prompt on InternVL2-8B. Results are summarized below.
>
> | Method | BLEU-1 | Avg. BLEU | ROUGE-1 | ROUGE-2 | ROUGE-L | METEOR |
> |---|---|---|---|---|---|---|
> | ICL | 30.2 | 17.5 | 35.2 | 13.4 | 28.7 | 31.8 |
> | ICL (with style and structure hint) | 29.4 | 17.9 | 34.0 | 12.5 | 27.7 | 30.8 |
>
> Adding the hint does not improve ICL performance and in fact, it leads to a marginal drop, likely because explicitly instructing the model to follow exemplar style and structure reduces attribute copying from exemplars, which in turn slightly lowers lexical overlap with the reference.

---

> > ### Author Response · Authors · 2026-04-21
> > **Author Response to Reviewer 1k3U - 2**
> >
> > **Comment-4**: The paper tests and uses K=1, while methods for comparisons still seem to use K=2. Would these differences lead to unfair comparisons?
> >
> > **Resp.**: Thank you for raising this concern. Apologies, we conflated two distinct roles of the parameter K under the same notation in the original manuscript. (1) K for Chain-of-Attributes controls how many retrieved products are used to aggregate the attribute key set (Section 3.2.2). Based on our ablation (Table 2), we set K=1 here, as a compact set of high-confidence attributes from the single closest retrieved sample provides a more favorable balance between attribute relevance and captioning accuracy than larger sets at K=2 or K=3. (2) K for ICL exemplars controls how many retrieved products are provided as exemplars during caption generation (Section 3.2.4), and is set to K=2 consistently across both RA-CoA and the ICL baselines, ensuring a fair comparison. **We have revised the manuscript to use distinct notation (e.g., K for attribute key set and K_icl for ICL exemplars) to make this distinction explicit.**
> >
> > **Comment-5**: Third, the generalization experiment is conducted on a dataset with merely 50 images. How could this small-scale dataset represent the diversity of real-world data?
> >
> > **Resp.**: We thank the reviewer for pointing this out. To thoroughly validate the generalizability of our approach to real-world e-commerce data, we expand our manually-curated evaluation set from 50 to 500 samples. We specifically focus on products recently added to e-commerce platforms, to ensure no train-test leakage and hence a fair comparison with the fully supervised baseline UniFashion. Our new evaluation data spans around 17 product categories, with an average of 28.5 products per category. The gender distribution of products between male, female and unisex is 40%, 48% and 12% respectively. The products capture a variety of styles including casual, formal, activewear, partywear, etc.
> >
> > | Method | BLEU-1 | Avg. BLEU | ROUGE-1 | ROUGE-2 | ROUGE-L | METEOR |
> > |---|---|---|---|---|---|---|
> > | UniFashion | 11.36 | 3.91 | 16.46 | 4.11 | 12.48 | 16.20 |
> > | **Ours** | **21.98** | **7.69** | **26.05** | **5.66** | **18.34** | **20.21** |
> >
> > We observe a consistent performance improvement with RACoA across all metrics, thereby showcasing better real-world generalization of RA-CoA on a larger evaluation set despite being completely training-free. **We have updated these results in Table 8 of the manuscript.**
> >
> > **Comment-6**: Noisy Knowledge Base paragraph, how does this experiment of adding noise value verify the choice of using attributes instead of values? Where are the results of using attributes only and using attributes+values?
> >
> > **Resp.**: The attribute-only choice for CoA was motivated by the observation that similar products share attribute types but differ in specific values. For example, a round-neck T-shirt may be retrieved for a V-neck query: the key "neckline" is shared but values differ. Providing retrieved values in CoA introduces two problems: (i) when K retrieved products carry different values for the same attribute, supplying all of them introduces conflicting signals while selecting one arbitrarily introduces selection bias; (ii) even a single retrieved value increases memorization bias, encouraging the model to copy rather than ground the value in the query image. The keys-only design retains attribute guidance while leaving value prediction entirely to the VLM's visual grounding.
> >
> > The noisy KB experiment (Table 6) was designed to analyze robustness to corrupted KB entries. Since values never enter the CoA reasoning chain, corrupted values propagate only to exemplar captions used during caption generation. Table 6 therefore quantifies the performance drop attributable to degraded exemplar quality, and the limited degradation confirms robustness to noisy KB entries.
> >
> > **Comment-7**: While including a user study is appreciated, the paper needs to convince others that the selected six evaluators represent diverse users.
> >
> > **Resp.**: We thank the reviewer for this observation. Our six evaluators were recruited with attention to diversity across gender, age, and regional background, two female and four male participants, spanning two age groups (18-24 and 25-30, three each), drawn from four distinct regions with varied linguistic and cultural backgrounds. All evaluators were proficient English speakers with prior experience using fashion e-commerce platforms. Further, the following two design choices mitigate the modest panel size. First, each evaluator rated 50 disjoint randomly sampled test images, yielding 300 unique evaluation instances that substantially reduce individual-level bias. Second, the preference margin for RA-CoA is large (68.6%) and consistent across evaluators from different demographic subgroups, suggesting the finding is not driven by any particular evaluator profile.

---

> > > ### Author Response · Authors · 2026-04-21
> > > **Author Response to Reviewer 1k3U - 3**
> > >
> > > **Comment-8**: According to Table 5, the results of 40K are very close (better in two metrics, while lower in four with 0.1 or 0.2) to those of 60K. The paper needs to explain why an additional 20K is necessary.
> > >
> > > **Resp.**: While the performance gap between 40K and 60K is marginal on average metrics, a larger ProductKB ensures broader coverage of diverse fashion categories, styles, and attribute vocabularies, which is critical for rare or niche products that may not be well-represented in smaller KBs. Average metrics over the full test set may not fully reflect improvements on such long-tail items, where retrieval quality depends directly on KB diversity and scale. **We have added this clarification to the Section 4.7.5 of the manuscript.**
> > >
> > > **Comment-9**: Caption Generation with Oracle Attributes paragraph, I am not sure where to obtain the ground truth attribute keys.
> > >
> > > **Resp.**: As stated in Section 4.7.3, since FashionGen does not provide ground-truth attribute-value pairs directly, we derive them from the human-expert-written captions using the same LLaMA-3.2-3B-Instruct extraction pipeline used for ProductKB construction (Section 3.1). The extracted attribute keys from these captions serve as the oracle inputs.
> > >
> > > **Comment-10**: To strengthen the relevance to TMLR's readership, the paper needs to highlight its insights behind a simple combination of RAG and CoT. Fine-grained attributes could be one point, yet whether this method can generalize to other tasks that require understanding fine-grained attributes is currently unknown.
> > >
> > > **Resp.**: We thank the reviewer for this suggestion. RA-CoA's insights go beyond a simple combination of RAG and CoT. The non-obvious design choices that distinguish it are: (i) retrieving only attribute keys from similar products, not their values, which prevents propagation of incorrect values while constraining the reasoning space to domain-relevant attributes; (ii) product-aware cropping via Florence-2 for retrieval, which ensures retrieved samples are semantically aligned to the target product; and (iii) sequential per-attribute querying via CoA, which reduces cognitive load on the VLM by isolating one attribute at a time rather than asking for free-form reasoning over the full attribute space. The noisy KB experiment (Table 6) empirically validates the keys-only design: corrupted attribute values in the KB cause only marginal performance degradation precisely because values never enter the CoA reasoning chain.
> > >
> > > Further, **to assess generalizability to other fine-grained product domains**, we evaluated RA-CoA on a furniture dataset [1] with structured attribute annotations. Unlike FashionGen where captions are available, this dataset provides ground truth attribute-value pairs directly; we generate natural language captions from these ground truth attribute-value pairs using an LLM (Qwen3-8B). The training split is used as the KB and evaluation is performed on the test split. The following results are obtained on this dataset:
> > >
> > > | Method | BLEU-1 | Avg. BLEU | ROUGE-1 | ROUGE-2 | ROUGE-L | METEOR | LLM-Judge |
> > > |---|---|---|---|---|---|---|---|
> > > | ZS | 4.0 | 1.4 | 22.8 | 2.0 | 17.4 | 9.2 | 65.7 |
> > > | COT | 38.6 | 17.7 | 36.5 | 11.2 | 28.3 | 27.4 | 75.1 |
> > > | ICL | 71.0 | 55.0 | 68.2 | 44.8 | 67.5 | 72.8 | 34.2 |
> > > | RA-CoA | **74.5** | **60.6** | **74.8** | **51.3** | **71.4** | **73.2** | **77.2** |
> > >
> > > RA-CoA outperforms all baselines across all metrics. Two patterns are worth noting. First, CoT achieves a strong LLM-judge score (75.1) despite lower lexical scores, suggesting that furniture attributes are visually salient enough for free-form reasoning to identify them correctly, but CoT lacks the exemplar-guided structure to produce well-formed captions. Second, ICL achieves high lexical scores but a dramatically lower LLM-judge score (34.2), consistent with exemplar style copying that inflates n-gram overlap without improving attribute correctness. RA-CoA resolves both limitations: CoA ensures accurate attribute grounding while retrieved exemplars guide caption structure, achieving the highest scores across all metrics. **We have added this result and the discussion in the Section 4.7.7 of the updated manuscript.**
> > >
> > > [1] Nowicki, Filip, Arkadiusz Charliński, and Andrzej Wójtowicz. "Future Designer-Generative AI Meets Interior Design." Joint European Conference on Machine Learning and Knowledge Discovery in Databases. Cham: Springer Nature Switzerland, 2025.

---

### Review · Reviewer_cuu5 · 2026-04-08

**Summary Of Contributions:**

This paper proposes RA-CoA, a training-free framework for fashion image captioning that combines retrieval-augmented knowledge with structured attribute-level reasoning. The method decomposes caption generation into (i) retrieval of attribute schemas from a Product Knowledge Base (ProductKB), and (ii) chain-of-attribute (CoA) reasoning, where a vision-language model predicts attribute values sequentially before synthesizing a final caption. The approach aims to address limitations of zero-shot vision-language models in capturing fine-grained fashion attributes. Empirical results demonstrate consistent improvements across multiple VLMs. The method is practical and well-motivated for real-world e-commerce applications. However, several aspects of the experimental design and evaluation limit the strength of the conclusions.

**Audience:**

Yes

**Audience Explanation:**

The paper combines retrieval augmentation with chain of attributes. While the focus is specifically on fashion image captioning, the framework can be tested in other domains, such as furniture, electronics, etc.

**Broader Impact Concerns:**

An ethical consideration statement is available.

**Claims And Evidence:**

Yes

**Claims Explanation:**

The ablation studies are thorough. However, further evaluation is required to disentangle individual contributions.

**Requested Changes:**

1)	How does performance change when using retrieval-based ICL instead of random ICL discussed in section 4.2.1? The observed gains over ICL may partly reflect the benefit of smarter exemplar selection rather than the chain-of-attribute mechanism itself.
2)	Given that cropping, intended to improve retrieval quality, yields only modest gains over full-image inputs (Table 4), how much does the retrieval step itself contribute beyond direct prompting with a strong VLM?
3)	Can the method handle novel attributes not present in ProductKB? The paper does not address how RA-CoA behaves when a query product contains attributes genuinely absent from the ProductKB. It is a realistic scenario as fashion inventories evolve.
4)	The paper does not disentangle the contributions of retrieval and chain-of-attribute reasoning. Since RA-CoA combines both components, it is unclear how much each contributes to the observed improvements. Including ablations with retrieval-only and CoA-only variants would be necessary to isolate their individual impact.

Minor Recommendation

1)	The real-world generalization comparison (Table 8) is limited to a single supervised baseline (UniFashion) on only 50 images. Can the authors evaluate against additional supervised SOTA methods on a larger held-out set to strengthen this comparison?

2)	Does the framework generalize to other fine-grained product domains such as electronics or furniture, where attribute-level captioning is equally critical?

3)	Consider code and model release for reproducibility

---

> ### Author Response · Authors · 2026-04-21
> **Author response to Reviewer cuu5**
>
> **Comment-1**: How does performance change when using retrieval-based ICL instead of random ICL discussed in section 4.2.1? The observed gains over ICL may partly reflect the benefit of smarter exemplar selection rather than the chain-of-attribute mechanism itself.
>
> **Comment-2**: Given that cropping, intended to improve retrieval quality, yields only modest gains over full-image inputs (Table 4), how much does the retrieval step itself contribute beyond direct prompting with a strong VLM?
>
> **Comment-4**: The paper does not disentangle the contributions of retrieval and chain-of-attribute reasoning. Since RA-CoA combines both components, it is unclear how much each contributes to the observed improvements. Including ablations with retrieval-only and CoA-only variants would be necessary to isolate their individual impact.
>
> **Common Response for Comment-1/Comment-2/Comment-4**: We thank the reviewer for these observations, which collectively ask how much each component of RA-CoA independently contributes. We address this with two complementary analyses.
>
> *(i) Retrieval vs. CoA contribution*: To demonstrate that CoA contributes independently beyond exemplar selection quality, we evaluated retrieval-based ICL on InternVL2-8B. Results are summarized below.
>
> | Method | BLEU-1 | Avg. BLEU | ROUGE-1 | ROUGE-2 | ROUGE-L | METEOR |
> |---|---|---|---|---|---|---|
> | ICL (random exemplars) | 30.2 | 17.5 | 35.2 | 13.4 | 28.7 | 31.8 |
> | ICL (retrieved exemplars) | 31.1 | 21.2 | 40.9 | **20.3** | 35.1 | 34.9 |
> | RA-CoA | **38.6** | **23.7** | **41.0** | 17.4 | **36.0** | **38.1** |
>
> RA-CoA outperforms retrieval-based ICL across primary metrics: METEOR 38.1 vs 34.9, BLEU-1 38.6 vs 31.1, and Avg. BLEU 23.7 vs 21.2, confirming that CoA contributes independently beyond smarter exemplar selection.
>
> *(ii) Role of ICL exemplars vs. CoA*: ICL exemplars primarily drive the structure and style of the generated caption. As shown in Table 3, RA-CoA with CoA mechanism alone, without any exemplar guidance, achieves METEOR 19.8, outperforming all non-exemplar baselines: zero-shot (METEOR 13.8), CoT-i (15.7), and CoT-e (13.5) on InternVL2-8B. This demonstrates that CoA independently contributes to accurate attribute identification and grounding directly from the image. When CoA-derived attribute-value pairs are subsequently combined with retrieved ICL exemplars during caption generation, performance improves further to 38.1 METEOR, as the exemplars guide caption structure and style while CoA ensures attribute accuracy and interpretability. Retrieval and CoA are thus complementary and additive: retrieval improves exemplar quality and attribute key coverage, while CoA ensures accurate attribute value grounding in the query image.
>
> **Comment-3**: Can the method handle novel attributes not present in ProductKB? The paper does not address how RA-CoA behaves when a query product contains attributes genuinely absent from the ProductKB. It is a realistic scenario as fashion inventories evolve.
>
> **Resp.**: We thank the reviewer for raising this important and realistic scenario. In RA-CoA, caption generation is conditioned on the subset of attributes retrieved from the ProductKB. When a query product contains an attribute (say K1) that is absent from the ProductKB, the model relies on the remaining available (closest possible) attributes (say K2, ..., K5) to construct the caption (please note that having all attributes being unseen is a rare scenario, which we do not consider in this work). As a result, the generated caption remains semantically meaningful and partially informative, but may omit the missing attribute-value pair (K1→V1). This reflects a graceful degradation behavior rather than catastrophic failure.
>
> Importantly, RA-CoA does not explicitly enforce recovery of such unseen attribute keys, which we acknowledge as a limitation. However, for novel attribute values associated with keys that are retrievable, the VLM's broad parametric knowledge enables open-ended value inference beyond any closed vocabulary. For instance, given a product with a culturally specific print pattern such as Kalamkari or Ikat, once the key "print" is surfaced from a retrieved similar product, CoA queries the VLM: "what is the print of this product?" and the VLM's visual and linguistic pretraining allows it to generate the semantically correct value. Similarly, for products with printed text or logos, the VLM's internal OCR capabilities can accurately recover the value once the relevant key is provided, a capability that supervised methods with closed output vocabularies fundamentally lack.
>
> We agree that handling truly novel attribute keys absent from the KB remains an open challenge. A practical advantage of RA-CoA's interpretable design is that one can manually inspect and augment the CoA output with new key-value pairs at deployment time, without any retraining.

---

> > ### Author Response · Authors · 2026-04-21
> > **Author response to Reviewer cuu5 - 2**
> >
> > **Comment-5**: The real-world generalization comparison (Table 8) is limited to a single supervised baseline (UniFashion) on only 50 images. Can the authors evaluate against additional supervised SOTA methods on a larger held-out set to strengthen this comparison?
> >
> > **Resp.**: We thank the reviewer for this suggestion. UniFashion represents the current state-of-the-art in supervised fashion image captioning, having been extensively trained across seven fashion tasks on large-scale datasets including FashionGen and FashionIQ, making it the strongest available supervised baseline for this comparison. We expand our evaluation set from 50 to 500 manually curated samples, focusing on products recently added to e-commerce platforms to ensure no train-test leakage with the fully supervised UniFashion baseline. The expanded set spans 17 product categories with an average of 28.5 products per category, covering diverse styles including casual, formal, activewear, and partywear, with a gender distribution of 40% male, 48% female, and 12% unisex products, ensuring broad representativeness of real-world e-commerce data.
> >
> > | Method | BLEU-1 | Avg. BLEU | ROUGE-1 | ROUGE-2 | ROUGE-L | METEOR |
> > |---|---|---|---|---|---|---|
> > | UniFashion | 11.36 | 3.91 | 16.46 | 4.11 | 12.48 | 16.20 |
> > | **Ours** | **21.98** | **7.69** | **26.05** | **5.66** | **18.34** | **20.21** |
> >
> > RA-CoA outperforms UniFashion across all metrics on this larger evaluation set, with METEOR gains of 4.0 points and BLEU-1 gains of 10.6 points, further validating its real-world generalization despite being completely training-free. **We have updated these results in Table 8 of the manuscript.**
> >
> > **Comment-6**: Does the framework generalize to other fine-grained product domains such as electronics or furniture, where attribute-level captioning is equally critical?
> >
> > **Resp.**: **To assess generalizability to other fine-grained product domains**, we evaluated RA-CoA on a furniture e-commerce dataset with structured attribute annotations. Unlike FashionGen where captions are available, this dataset provides ground truth attribute-value pairs directly; we generate natural language captions from these ground truth attribute-value pairs using Qwen3-8B LLM. The training split is used as the KB and evaluation is performed on the test split.
> >
> > | Method | BLEU-1 | Avg. BLEU | ROUGE-1 | ROUGE-2 | ROUGE-L | METEOR | LLM-Judge |
> > |---|---|---|---|---|---|---|---|
> > | ZS | 4.0 | 1.4 | 22.8 | 2.0 | 17.4 | 9.2 | 65.7 |
> > | COT | 38.6 | 17.7 | 36.5 | 11.2 | 28.3 | 27.4 | 75.1 |
> > | ICL | 71.0 | 55.0 | 68.2 | 44.8 | 67.5 | 72.8 | 34.2 |
> > | RA-CoA | **74.5** | **60.6** | **74.8** | **51.3** | **71.4** | **73.2** | **77.2** |
> >
> > RA-CoA outperforms all baselines across all metrics. Two patterns are worth noting. First, CoT achieves a strong LLM-judge score (75.1) despite lower lexical scores, suggesting that furniture attributes are visually salient enough for free-form reasoning to identify them correctly, but CoT lacks the exemplar-guided structure to produce well-formed captions. Second, ICL achieves high lexical scores but a dramatically lower LLM-judge score (34.2), consistent with exemplar style copying that inflates n-gram overlap without improving attribute correctness. RA-CoA resolves both limitations: CoA ensures accurate attribute grounding while retrieved exemplars guides caption structure, achieving the highest scores across all metrics. **We have added this result and the discussion in the Section 4.7.7 of the updated manuscript.**
> >
> > **Comment-7**: Consider code and model release for reproducibility.
> >
> > **Resp**: We will release the code upon acceptance of this work.

---

### Review · Reviewer_tDgL · 2026-04-10

**Summary Of Contributions:**

This paper proposes RA-COA, a training-free framework that disentangles fashion image captioning into retrieval of relevant attribute sets from a product knowledge base and attribute-level reasoning to generate the final caption.  Extensive experiments across open-source and closed-source VLMs show large gains over zero-shot, ICL, and CoT prompting.

**Audience:**

Yes

**Audience Explanation:**

From application side, the fashion image captioning is very important in enhancing user experience and product search in e-commerce platforms. From the methodology side, the proposed RA-CoA is intuitive and leads to performance gain. I think it can also been used for other fine-grained image captioning tasks.

**Claims And Evidence:**

Yes

**Claims Explanation:**

The authors conduct extensive experiments aross several open-source and closed-source VLMs, and the results show significant improvements over zero-shot, ICL, and CoT prompting. The authors also conduct comprehensive ablation studies regarding the tretrieval size, KB noise, sparsity, cropping, which shows the effectiveness of these design choices. Also the author provide numerical evaluation of the latency. These numerical results support most of the authors claims.

**Requested Changes:**

1. There are many existing visual RAG baselines, such as [1]. The authors should compare RA-CoA with these baselines.
2. Another simple baseline is the combination of CoT and ICL.
3. I am curious on how large is the pool of attributes. Will there be any similar or redundant attributes in the KB? I think the total number of attributes for fashion is limited and can be enumerated, why not enumerate these attributes in the prompt?
4. The authors claim one challenge of fashion image captioning is that there will  emerging trends, seasonal styles, and new vocabularies. However, in the experiments, the dataset used to built the KB is an old dataset (2018), and the corresponding vocabularies should already be known by the modern VLMs. This claim are not supported by experiments.
5. The RAG used in this pipeline mainly focus on extraction of the attributes. This is more like to provide the model with a correct format, which will improve the metric such as BLEU and ROUGE. However, it does not have a direct effects to help the model to correctly fill in the attributes. This should be the most critic part of the task that will directly influence the search results and users' experience. Could the authors comment more on this?

[1] Bonomo, Mirco, and Simone Bianco. "Visual RAG: Expanding MLLM visual knowledge without fine-tuning." arXiv preprint arXiv:2501.10834 (2025).

---

> ### Author Response · Authors · 2026-04-21
> **Author response to Reviewer tDgL**
>
> **Comment-1**: There are many existing visual RAG baselines, such as [1]. The authors should compare RA-CoA with these baselines.
>
> **Resp.**: We thank the reviewer for this suggestion. **We have now added this paper in the related work.** Visual RAG proposes CLIP-based retrieval-augmented in-context learning for image classification; however, its direct applicability to fine-grained, open-ended attribute-level caption generation remains limited. While the task setup differs from open-ended attribute-level caption generation, we adapt a similar spirit by evaluating ICL with CLIP-based retrieval (without product-aware cropping) as an additional baseline on InternVL2-8B. Results are summarized below.
>
> | Method | BLEU-1 | Avg. BLEU | ROUGE-1 | ROUGE-2 | ROUGE-L | METEOR |
> |---|---|---|---|---|---|---|
> | Adapted Visual RAG | 24.2 | 16.6 | 38.2 | **18.7** | 32.4 | 29.4 |
> | **RA-CoA** | **38.6** | **23.7** | **41.0** | 17.4 | **36.0** | **38.1** |
>
> Adapted Visual RAG still substantially underperforms RA-CoA across primary metrics: METEOR 29.4 vs 38.1 and BLEU-1 24.2 vs 38.6, confirming that retrieval-augmented exemplar selection alone is insufficient for fine-grained attribute-level captioning, and that the structured per-attribute visual grounding provided by CoA is the key driver of RA-CoA's gains.
>
> **Comment-2**: Another simple baseline is the combination of CoT and ICL.
>
> **Resp.**: Thank you for this suggestion. We agree that ICL+CoT-e constitutes a stronger baseline, and we have added this experiment. The results are summarized in the table below.
>
> | Model | | BLEU-1 | Avg. BLEU | ROUGE-1 | ROUGE-2 | ROUGE-L | METEOR |
> |---|---|---|---|---|---|---|---|
> | Qwen-2.5VL (3B) | ICL+CoT-e | 9.0 | 3.1 | 20.5 | 3.9 | 15.0 | 16.3 |
> | | **Ours** | **26.8** | **14.5** | **35.3** | **11.7** | **28.3** | **28.6** |
> | Qwen-2.5VL (7B) | ICL+CoT-e | 10.6 | 4.0 | 23.1 | 5.4 | 17.6 | 16.5 |
> | | **Ours** | **31.4** | **17.7** | **39.2** | **14.2** | **32.3** | **32.4** |
> | InternVL2-8B | ICL+CoT-e | 14.2 | 7.1 | 23.4 | 7.5 | 19.0 | 24.8 |
> | | **Ours** | **38.6** | **23.7** | **41.0** | **17.4** | **36.0** | **38.1** |
>
> RA-CoA consistently outperforms ICL+CoT-e across all models and metrics. For instance, on InternVL2-8B, RA-CoA achieves a METEOR score of 38.1 against 24.8 for ICL+CoT-e, a gain of 13.3 points. We also note that ICL+CoT-e underperforms ICL alone, which is consistent with a trend already visible in Table 1: CoT-e underperforms even zero-shot for most open-source models, indicating that these models struggle to follow unstructured explicit reasoning instructions reliably. Adding ICL exemplars partially recovers this degradation, but the inherent weakness of unstructured explicit reasoning still limits overall performance. This further reinforces our core motivation: RA-CoA replaces free-form reasoning with focused, one-attribute-at-a-time queries via Chain-of-Attributes, providing the structured visual grounding that ICL+CoT-e lacks. **We have added this result in Table 1 of the manuscript.**
>
> **Comment-3**: I am curious on how large is the pool of attributes. Will there be any similar or redundant attributes in the KB? I think the total number of attributes for fashion is limited and can be enumerated, why not enumerate these attributes in the prompt?
>
> **Resp.**: Our ProductKB contains 9,596 unique attributes, with each product having 9 attributes on average. From the 9.6K unique attributes, the top 85 attributes (high-frequency ones like color, neckline, sleeve length, etc.) account for 80% of the attribute slots across 60K products. The coverage reaches 95% with top 731 attributes but the remaining long tail captures precisely the details that drive purchase decisions. Including only the frequent attributes would systematically miss the ones that differentiate products. For example, for a plain T-shirt an attribute like ‘sleeve_style’ is irrelevant and not generally included, but for a puff-sleeved top, sleeve_style is the single most distinguishing feature (USP) a customer notices and searches for. Retrieval adaptively selects the right schema per product. We also note that e-commerce attributes are seller-supplied, meaning naming conventions vary across vendors. A fixed enumeration would be brittle to this inconsistency; retrieval is naturally robust since it can adapt to existing annotation schema for particular e-commerce platforms.

---

> > ### Author Response · Authors · 2026-04-21
> > **Author response to Reviewer tDgL - 2**
> >
> > **Comment-4**: The authors claim one challenge of fashion image captioning is that there will emerging trends, seasonal styles, and new vocabularies. However, in the experiments, the dataset used to built the KB is an old dataset (2018), and the corresponding vocabularies should already be known by the modern VLMs. This claim are not supported by experiments.
> >
> > **Resp:** We thank the reviewer for this insightful observation. We agree that the dataset used to construct the knowledge base is from 2018, and therefore may not fully capture recent emerging trends or newly coined vocabulary. Our intention was not to claim that the dataset itself contains unseen vocabulary for modern VLMs, but rather to highlight a broader challenge in fashion captioning, that the domain is inherently dynamic, with frequent introduction of new styles and terminology, and re-training each time may not be possible.
> >
> > Our current experiments should be interpreted as a proof-of-concept demonstrating the utility of knowledge augmentation, rather than a direct evaluation on truly novel vocabulary. Please also refer to our response to Comment-3 by Reviewer cuu5.
> >
> > **Comment-5**: The RAG used in this pipeline mainly focus on extraction of the attributes. This is more like to provide the model with a correct format, which will improve the metric such as BLEU and ROUGE. However, it does not have a direct effects to help the model to correctly fill in the attributes. This should be the most critic part of the task that will directly influence the search results and users' experience. Could the authors comment more on this?
> >
> > **Resp:** We thank the reviewer for highlighting this important point. We agree that correctly identifying attribute values is the most critical aspect of fashion image captioning, as it directly impacts downstream search and user experience.
> >
> > We would like to clarify that the role of retrieval in RA-CoA is not merely to enforce a structured output format. Instead, it provides attribute-specific cues that guide the model’s reasoning. In the Chain-of-Attribute (CoA) step, decomposing the task into individual attributes allows the VLM to attend to relevant regions of the image for each attribute, reducing ambiguity and improving the accuracy of value prediction.
> >
> > Empirically, this effect goes beyond improvements in surface-form metrics such as BLEU or ROUGE. As shown in Table 1 (Section 4.4.2), RA-CoA also achieves higher scores on our LLM-as-Judge evaluation, which explicitly measures whether the generated captions correctly capture gold attribute values. This indicates that the gains stem from improved attribute correctness rather than formatting alone.

---

### Decision · Action_Editor_DF6v · 2026-05-18

**Recommendation:** Accept as is

**Audience:**

Yes

**Audience Explanation:**

This paper develops new algorithms for fashion image captioning, which is an interesting and practical domain.

**Claims And Evidence:**

Yes

**Claims Explanation:**

This paper proposes a training-free framework named RA-CoA for fashion image captioning. It combines retrieved product knowledge with structured attribute reasoning. The method has two main steps. First, it retrieves attribute schemas from a Product Knowledge Base, called ProductKB. Second, it uses chain-of-attribute reasoning, in which a vision-language model predicts attribute values one by one. It then uses these attributes to generate the final caption. The experiments show consistent gains across several VLMs. The method is practical and useful for real-world e-commerce applications.

Reviewers raised some concerns regarding experimental settings, baselines, model generalizability, paper presentation, etc. The authors have revised the paper substantially and provided detailed responses. Reviewers agreed that their previous concerns have been well addressed in the revised version. The claims made in the paper are supported by accurate, convincing, and clear evidence.